# Latent Particle World Models: Self-supervised Object-centric Stochastic Dynamics Modeling

**Tal Daniel**[1], **Carl Qi**[2], **Dan Haramati**[3], **Amir Zadeh**[4], **Chuan Li**[4], **Aviv Tamar**[5],
**Deepak Pathak**[1], **David Held**[1]
[1]Carnegie Mellon University, [2]UT Austin, [3]Brown University, [4]Lambda, [5]Technion
tdaniel@andrew.cmu.edu

## ABSTRACT

We introduce Latent Particle World Model (LPWM), a self-supervised object-centric world model scaled to real-world multi-object datasets and applicable in decision-making. LPWM autonomously discovers keypoints, bounding boxes, and object masks directly from video data, enabling it to learn rich scene decompositions without supervision. Our architecture is trained end-to-end purely from videos and supports flexible conditioning on actions, language, and image goals. LPWM models stochastic particle dynamics via a novel latent action module and achieves state-of-the-art results on diverse real-world and synthetic datasets. Beyond stochastic video modeling, LPWM is readily applicable to decision-making, including goal-conditioned imitation learning, as we demonstrate in the paper. Code, data, pre-trained models and video rollouts are available:
https://taldatech.github.io/lpwm-web

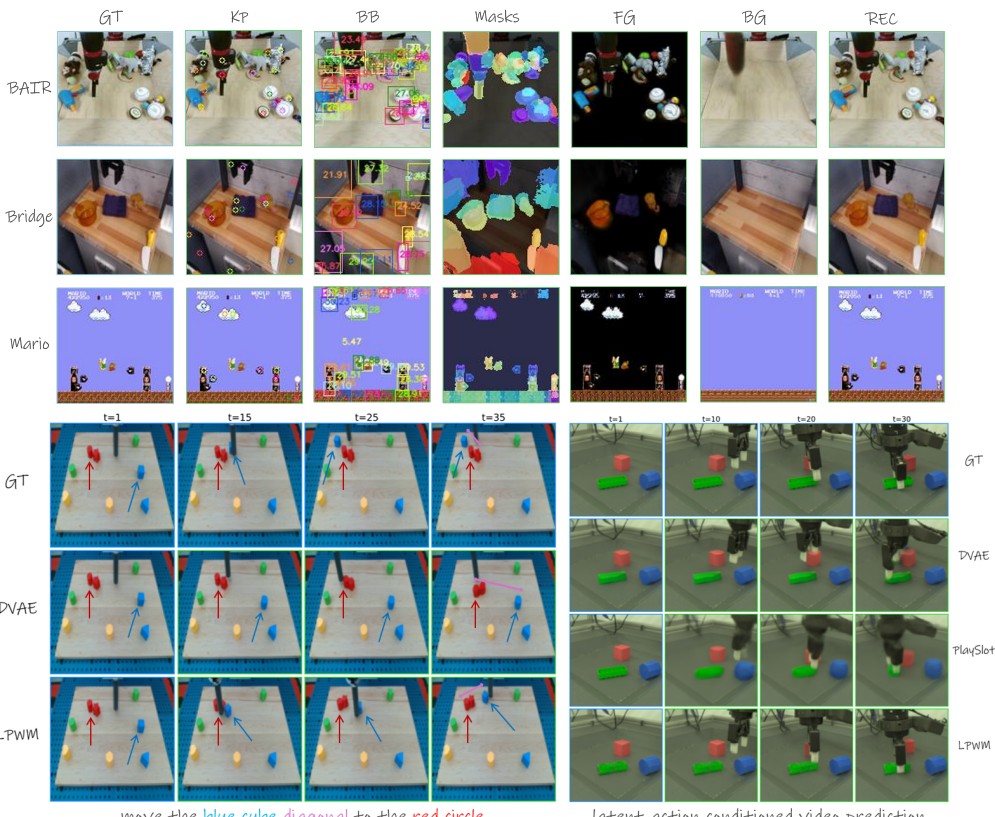

Figure 1: Self-supervised object-centric world modeling with LPWM. Top: latent particle decomposition. Bottom left: language-conditioned video generation. Bottom right: latent-action-conditioned video prediction.

# 1 INTRODUCTION

Recent years have witnessed remarkable progress in the visual fidelity of general-purpose video generation models (Blattmann et al., 2023; Yang et al., 2024b). Driven by vast datasets and expansive computational resources, these models—often built on scalable architectures like Transformers (Vaswani et al., 2017)—have achieved unprecedented realism. However, their success comes at a steep computational cost: training requires thousands of GPU hours (Zhu et al., 2024), and, due to their reliance on diffusion processes (Ho et al., 2020), inference remains slow and resource-intensive, limiting practical applications. This has sparked an important question: can we leverage the strengths of these generative models for decision-making? For instance, by turning them into *world models*—dynamics predictors that can be externally controlled by action or goal signals, tasks such as robotic planning (Yang et al., 2023; Zhu et al., 2024) become possible. Despite their strengths in producing high-fidelity videos, these models' resource demands can be prohibitive. In parallel, recent work (Haramati et al., 2024; Qi et al., 2025) demonstrates that incorporating inductive biases and leveraging more compact models can enable efficient, robust performance on complex multi-object decision-making tasks. Motivated by this, our work aims to bridge these two directions—by introducing an efficient, self-supervised, object-centric world model for video prediction, *and* decision-making in real-world and simulated, multi-entity environments.

Consider the illustrative example in Figure 2, where the dynamics of two moving objects are described alongside the caption: "The blue ball is moving diagonally towards the green square." Text representations typically rely on semantic tokenization into words or subword units, which underpins the success of large language models (Radford et al., 2018). In contrast, image representations predominantly use "patchifying"—dividing the image into a fixed grid of patches without regard for semantic content (Dosovitskiy et al., 2020). While this patch-based approach enables scalability and generality, it lacks the semantic intuitiveness of object-centric decompositions that can enhance the model's ability to capture meaningful object interactions and relationships, crucial for understanding complex scenes, and align more naturally with language representations. Inspired by the "what-where" pathway in the human visual system (Goodale & Milner, 1992) and recent neuroscience findings (Pickering & Clark, 2014; Nau et al., 2018; Barnaveli et al., 2025), which suggest humans leverage internal visual-spatial world models for planning and action, combining object-centric representations with world models is a promising direction towards more effective decision-making and vision-language integration.

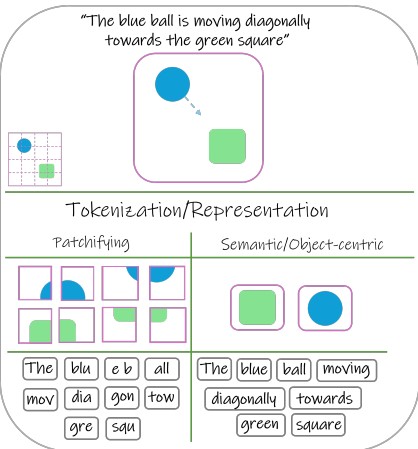

Figure 2: Representation discrepancy. Text is typically tokenized into semantically meaningful units such as words or subwords, whereas image representations are most often constructed by dividing the image into a fixed grid of patches ("patchifying") that do not explicitly encode semantic content.

Building on this premise, recent research has focused on introducing inductive biases in the form of object-centric representations, namely Deep Latent Particles (DLP, Daniel & Tamar (2022a)), which have shown strong empirical benefits across a range of domains—including video prediction (Daniel & Tamar, 2024), reinforcement learning (RL) (Haramati et al., 2024), imitation learning (Qi et al., 2025), and microscopy (Goldenberg et al., 2025). These approaches demonstrate that, when applicable, object-centric representations can lead to improved downstream performance, even with smaller model sizes and less data. However, their success has so far been largely confined to specific datasets and environments, typically involving simulated scenes or simple real-world settings with isolated objects, limited camera motion, or single-agent interactions. Scaling object-centric models to handle the complexity of real-world multi-object environments remains a substantial challenge. While patch-based representations remain dominant for large-scale, general-purpose visual modeling, ongoing advances indicate that object-centric approaches offer clear advantages for decision-making tasks whenever the problem structure allows. The present work aims to advance this direction by developing scalable, efficient world modeling grounded in object-centric decomposition.

In this work, we introduce the Latent Particle World Model (LPWM), the first self-supervised object-centric world model capable of end-to-end training on complex real-world video data. Building upon the DLP-based video prediction framework DDLP (Daniel & Tamar, 2024), we eliminate the requirement for explicit particle tracking and propose a novel context module that predicts distributions over latent actions for each particle. This approach enables stochastic dynamics sampling and enables scalability to complex environments. LPWM is trained exclusively from video observations and supports optional conditioning on actions, language, images, and multi-view inputs. Its design can accommodate a range of decision-making applications, including unconditional video prediction pretraining and goal-conditioned imitation learning. By integrating object-centric representations with scalable stochastic dynamics modeling, LPWM advances the development of efficient and interpretable visual world models.

Our contributions are summarized as follows: (1) We propose a self-supervised object-centric world model with a novel latent action module that supports multiple conditioning types, including actions, language, images, and multi-view inputs; (2) We achieve state-of-the-art performance in object-centric video prediction on diverse real-world and simulated multi-object datasets, and (3) We demonstrate the applicability of our model to imitation learning on two complex multi-object environments, highlighting its utility for decision-making tasks.

## 2 RELATED WORK

This section provides an overview of related literature relevant to latent object-centric video prediction and world modeling. To the best of our knowledge, LPWM is the first self-supervised object-centric model that can be trained solely from videos, supports multi-view training, and enables diverse conditioning modalities, including actions, language, and goal images. Since no existing method shares this unique combination of capabilities, we briefly review several adjacent and complementary lines of work to highlight the context and novelty of our contributions. A more detailed survey of keypoints, latent actions, and decision-making methods is provided in Appendix A.6.

**General video prediction and latent world models:** Classic approaches encode images into latent spaces and predict future states with recurrent dynamics, often using convolutional encoders and RNNs (Finn et al., 2016a; Ha & Schmidhuber, 2018). Recent work has improved long-horizon prediction with discrete latent variables (Hafner et al., 2020b), hierarchical architectures (Wang et al., 2022), self-attention (Micheli et al., 2024), and language conditioning (Nematollahi et al., 2025), but most methods model frames holistically and lack explicit object decomposition, resulting in blurry or unstable predictions. Video diffusion models (Zhu et al., 2024) achieve high fidelity, but remain computationally intensive and do not incorporate object-centric biases.

**Unsupervised object-centric latent video prediction and world models:** Unsupervised object-centric video prediction methods learn latent dynamics on decomposed scene elements, typically using patch-, slot-, or particle-based representations. **Patch-based approaches** (e.g., G-SWM (Lin et al., 2020a)) represent objects using local latent attributes and typically model joint dynamics with RNNs and interaction modules. These methods rely on post-hoc matching object proposals across frames for temporal consistency, which—combined with unordered object representations—limits their scalability to complex or real-world video datasets. **Slot-based approaches** (Locatello et al., 2020) typically represent scenes as a set of slots: permutation-invariant latent vectors encoding spatial and appearance information for objects. These approaches generally adopt a two-stage training strategy: a slot decomposition is first learned independently, followed by a separate dynamics model trained on the inferred slots using RNNs (Nakano et al., 2023) or Transformers (Wu et al., 2022b; Villar-Corrales et al., 2023). In practice, slot-based methods suffer from inconsistent decompositions, blurry predictions, and convergence issues (Seitzer et al., 2023). **Particle-based approaches**, introduces as DLP (Daniel & Tamar, 2022a), provide compact, interpretable object representations using keypoint-based latent particles with extended attributes. DDLP (Daniel & Tamar, 2024) jointly trains a Transformer dynamics model and the particle representation, allowing stable object-centric decomposition and improved modeling of complex scenes. However, DDLP relies on particle tracking and sequential encoding, which restricts parallelization and stochasticity. Our proposed LPWM model is a direct extension to this lineage. LPWM eliminates the need for explicit tracking, enabling parallel encoding of all frames, trains end-to-end, and integrates a latent action distribution for stochastic world modeling. This allows the model to capture transitions such

as object occlusion, appearance, or random movements (e.g., agents or grippers), and supports comprehensive conditioning via actions, language, or goal images–advancing particle-based modeling to the world model regime and addressing unsolved limitations of previous work.

**Video prediction and world models with latent actions:** Several recent works have introduced *latent actions*—global latent variables representing transitions between consecutive frames—to learn controllable or playable environments from videos. Models like CADDY (Menapace et al., 2021) and Genie (Bruce et al., 2024) learn discrete latent actions by quantizing inverse module outputs, and conditioning dynamics on these codes in a two-stage training scheme. AdaWorld (Gao et al., 2025) proposes a continuous latent action space with strong KL regularization. PlaySlot (Villar-Corrales & Behnke, 2025) augments slot-based object-centric video prediction with discrete latent action conditioning, demonstrating benefits of object-level decomposition for controllable modeling. In contrast, our particle-based LPWM learns continuous, per-particle latent actions end-to-end with dynamics, naturally capturing stochastic multi-object interactions. LPWM's learned latent policy enables sampling latent actions during inference without external input, supporting stochastic video generation. It also supports diverse conditioning—including goal-conditioning—making it well-suited for post-hoc policy learning and control, as demonstrated in our experiments.

Table 1 summarizes key differences between self-supervised object-centric video prediction and world modeling methods.

| Model | Obj.-Centric Rep. | Latent Actions | Action Cond. | Text Cond. | End-to-End | Dyn. Module |
|---|---|---|---|---|---|---|
| SCALOR (Jiang et al., 2019) | Patch | – | – | – | ✓ | RNN |
| G-SWM (Lin et al., 2020a) | Patch | – | – | – | ✓ | RNN |
| STOVE (Kossen et al., 2019) | Patch | – | – | – | ✓ | RNN |
| OCVT (Wu et al., 2021b) | Patch | – | – | – | – | Transformer |
| GATSBI (Min et al., 2021) | Patch+Keypt | – | ✓ | – | ✓ | RNN |
| PARTS (Zoran et al., 2021) | Slots | – | – | – | ✓ | RNN |
| STEDIE (Nakano et al., 2023) | Slots | – | – | – | ✓ | RNN |
| SlotFormer (Wu et al., 2022b) | Slots | – | – | – | – | Transformer |
| OCVP (Villar-Corrales et al., 2023) | Slots | – | – | – | – | Transformer |
| TextOCVP (Villar-Corrales et al., 2025) | Slots | – | – | ✓ | – | Transformer |
| SOLD (Mosbach et al., 2024) | Slots | – | ✓ | – | – | Transformer |
| PlaySlot (Villar-Corrales & Behnke, 2025) | Slots | Discrete | – | – | – | Transformer |
| DLP (Daniel & Tamar, 2022a) | Particles | – | – | – | – | GNN |
| DDLP (Daniel & Tamar, 2024) | Particles | – | – | – | ✓ | Transformer |
| LPWM (Ours) | Particles | Cont. (per) | ✓ | ✓ | ✓ | Transformer |

Table 1: Comparison of object-centric video prediction and world modeling methods across key dimensions and representation types. Please refer to Table 4 for an extended comparison.

# 3 BACKGROUND

**Variational Autoencoders (VAEs, (Kingma & Welling, 2014)):** VAEs are likelihood-based latent variable models that maximize the evidence lower bound (ELBO) on the data log-likelihood:

$$\log p_\theta(x) \geq \mathbb{E}_{q_\phi(z|x)}\left[\log p_\theta(x|z)\right] - KL\big(q_\phi(z|x)\|p(z)\big) \equiv ELBO(x),$$

where $q_\phi(z|x)$ (the *encoder*) approximates the intractable posterior, and $p_\theta(x|z)$ (the *decoder*) models the likelihood. Typically, $q_\phi$, $p_\theta$, and the prior $p(z)$ are Gaussian distributions, enabling efficient training via the *reparameterization trick*. Minimizing the negative ELBO decomposes into a reconstruction loss and a KL regularization term.

Temporal VAEs (Lee et al., 2018; Ha & Schmidhuber, 2018) extend this framework to sequential data (e.g., videos) by training to maximize the sum of ELBOs over timesteps. Here, the prior for each latent at timestep $t$ is conditioned on previous latents through a dynamics model, i.e., $\sum_t KL\big(q_\phi(z^t|x_t)\|p_\xi(z^t|z^{<t})\big)$. This enables learning temporally coherent latent dynamics suitable for video prediction.

**Deep Latent Particles (DLP, Daniel & Tamar (2022a; 2024)):** a VAE-based self-supervised object-centric representation for images. Each image is modeled as a set of $M$ foreground latent particles alongside a single background particle. A foreground particle is defined as $z_{\text{fg}} = [z_p, z_s, z_d, z_t, z_f] \in \mathbb{R}^{6+d_{\text{obj}}}$, where each component encodes a disentangled stochastic attribute: position $z_p \sim \mathcal{N}(\mu_p, \sigma_p^2) \in \mathbb{R}^2$, representing the 2D keypoint coordinates; scale $z_s \sim \mathcal{N}(\mu_s, \sigma_s^2) \in \mathbb{R}^2$, representing bounding-box size; depth $z_d \sim \mathcal{N}(\mu_d, \sigma_d^2) \in \mathbb{R}$, specifying compositing order (indicating which particles appear in front of others within the rendered scene); transparency

$z_t \sim \text{Beta}(a, b) \in [0, 1]$, controlling visibility; and visual features $z_f \sim \mathcal{N}(\mu_f, \sigma_f^2) \in \mathbb{R}^{d_{\text{obj}}}$, encoding appearance of the local region around the particle. The particle attributes are illustrated in Figure 8 in the Appendix. The background is represented by a single particle $z_{\text{bg}} \sim \mathcal{N}(\mu_{\text{bg}}, \sigma_{\text{bg}}^2) \in \mathbb{R}^{d_{\text{bg}}}$, fixed at the image center and modeling background visual features. DLP additionally learns an alpha channel mask per particle as part of reconstruction, enabling pixel-space foreground-background decomposition. For a detailed overview of DLP and its components, as well as improvements introduced in this work, see Appendix A.3. Figure 1 presents example decompositions of DLP on various datasets used in this work.

**Notations:** Non-latent (observed) temporal variables are denoted with subscripts indicating the temporal index, e.g., $x_t$, while latent variables use superscripts, e.g., $z^t$. For latent particles, the superscript index $m$ denotes the particle number within the set, and subscripts refer to attribute types. For example, $z_p^{m,t}$ denotes the position (keypoint) attribute $p$ of particle $m$ at timestep $t$, whereas $z_{\text{bg}}^t$ represents the background particle at timestep $t$.

## 4 LATENT PARTICLE WORLD MODELS (LPWM)

Our objective is to construct a *world model*—a dynamics model $\mathcal{F}(I_{0:T-1}, c) = \hat{I}_{T:T+\tau-1}$ that, given a sequence of $T$ image observations $I_{0:T-1} \in \mathbb{R}^{T \times C \times H \times W}$ (where $C$ is the number of channels, $H$ and $W$ are image height and width), and *optionally* conditioning inputs $c$ (actions, language, etc.), generates a rollout of future predictions $\hat{I}_{T:T+\tau-1} \in \mathbb{R}^{\tau \times C \times H \times W}$ in an autoregressive manner. As modeling directly in pixel space is high-dimensional and sample inefficient, we propose an end-to-end latent world model, Latent Particle World Models (LPWM), which combines a compact self-supervised object-centric latent representation, based on DLP, with a novel learned dynamics module that operates over particle latents. The model is trained end-to-end such that *the representation is trained to be predictable* by the dynamics module.

The **Latent Particle World Model (LPWM)** consists of four components, jointly trained end-to-end as a VAE: the ENCODER $\mathcal{E}_\phi$, the DECODER $\mathcal{D}_\theta$, the CONTEXT $\mathcal{K}_\psi$ and the DYNAMICS $\mathcal{F}_\xi$. The pipeline, illustrated in Figure 3, proceeds as follows: input frames are encoded into particle sets by the ENCODER, decoded back to images by the DECODER for reconstruction loss, then processed by the CONTEXT module to sample latent actions, which are combined with particles in the DYNAMICS module to predict next-step particles states and compute per-particle KL. Below, we summarize the role of each module and describe, in the main text, the core novel contributions-particularly the context and dynamics modules. For completeness, extended component details, minor implementation modifications of DLP and design choices are provided in Appendix A.4.

**ENCODER $\mathcal{E}_\phi$ (Appendix A.4.1):** corresponds to the VAE's approximate posterior $q_\phi(z|x)$. It takes as input an image frame and outputs a set of latent particles: $\mathcal{E}_\phi(x = I_t) = [\{z_{\text{fg}}^{m,t}\}_{m=0}^{M-1}, z_{\text{bg}}^t]$. Each frame $I_t$ is represented by $M$ foreground latent particles $\{z_{\text{fg}}^{m,t}\}_{m=0}^{M-1}$, where each particle originates from per-patch learned keypoint (see Appendix A.4.1), and one background particle $z_{\text{bg}}^t$. Unlike DDLP, particle filtering to a subset $L \leq M$ is deferred to the decoder to preserve particle identities and eliminates the need for explicit tracking, a requirement in DDLP that necessitated sequential frame encoding. In contrast, the proposed approach enables encoding all frames in parallel. Foreground particles are parameterized as $z_{\text{fg}}^m \in \mathbb{R}^{6+d_{\text{obj}}}$, where the first six dimensions capture explicit attributes (e.g., spatial coordinates, scale, transparency), and the remaining $d_{\text{obj}}$ dimensions represent appearance features as described in Section 3. The background particle is defined as $z_{\text{bg}} \in \mathbb{R}^{d_{\text{bg}}}$, where $d_{\text{bg}}$ denotes the latent dimension of the background visual features. These features are encoded from a masked version of the original image, in which regions corresponding to visible foreground particles are masked out, as illustrated in Figure 9 in the Appendix.

**DECODER $\mathcal{D}_\theta$ (Appendix A.4.2):** corresponds to the VAE's likelihood $p_\theta(x|z)$. It takes as input a set of $L \leq M$ foreground particles together with a background particle, and reconstructs an image frame: $\mathcal{D}_\theta([\{z_{\text{fg}}^{l,t}\}_{l=0}^{L-1}, z_{\text{bg}}^t]) = \hat{I}_t$. Here, $L$ can be less than $M$ to allow particle filtering before rendering, based on transparency or confidence measures (Daniel & Tamar, 2024), reducing memory usage without compromising reconstruction quality. Each particle is decoded independently into an RGBA (RGB + Alpha channel) glimpse $\tilde{x}_l^p \in \mathbb{R}^{S \times S \times 4}$, where $S$ is the glimpse size, representing the reconstructed appearance of particle $l$. The alpha mask (Alpha channel) is modulated by the transparency and depth attribute of each particle, and the decoded glimpse is then placed into the full-

resolution canvas to create $\hat{x}_{\mathrm{fg}}$. The background is decoded from $z_{\mathrm{bg}}$ using a standard upsampling network to produce $\hat{x}_{\mathrm{bg}}$, and the final reconstructed image is stitched according to $\hat{x} = \alpha \odot \hat{x}_{\mathrm{fg}} + (1 - \alpha) \odot \hat{x}_{\mathrm{bg}}$, where $\alpha$ is the effective mask obtained from the compositing process.

**CONTEXT $\mathcal{K}_\psi$:** We now present the main novel component added to the DLP framework—the CONTEXT module $\mathcal{K}_\psi$—designed to model *stochastic dynamics* in actionless videos. In such videos, scene dynamics are not fully determined by initial frames (e.g., a ball beginning to roll (Lin et al., 2020a) where initial conditions fully determine future dynamics) but can also be influenced by external signals such as actions (e.g., a robotic gripper (Ebert et al., 2017a)).

Commonly, stochastic transitions are captured by introducing *latent actions* (Menapace et al., 2021; Bruce et al., 2024; Gao et al., 2025; Villar-Corrales & Behnke, 2025). Typically, a latent action $z_c$ is learned through an autoencoding scheme: an inverse model infers $z_c^t = \mathcal{K}_\psi^{\mathrm{inv}}(I_{t+1}, I_t)$ from consecutive frames, and a decoder reconstructs the future frame $\hat{I}_{t+1} = \mathcal{D}_\theta(I_t, z_c^t)$, trained via reconstruction loss. To prevent $z_c^t$ from trivially memorizing $I_{t+1}$, strong regularization is applied through a vector quantization bottleneck (Bruce et al., 2024; Ye et al., 2025) or KL-regularization to a fixed prior (Gao et al., 2025). However, these approaches use a *global* latent action vector representing all changes between frames, limiting their ability to model local dynamics in multi-entity scenes (e.g., independent enemy movements in `Mario` or secondary contact events in robotics). A global action vector cannot naturally disentangle these local dynamics.

In this work, we introduce the CONTEXT module $\mathcal{K}_\psi$, a novel *per-particle* mechanism for latent action modeling. Unlike prior work (Villar-Corrales & Behnke, 2025; Gao et al., 2025), we model a latent action for each particle, directly governing the transition from $z_i^{m,t}$ to $z_i^{m,t+1}$. Regularization is not imposed via a fixed prior, but instead learned through a *latent policy*, which models the distribution of latent actions conditioned on the current state. This per-particle formulation enables the representation of multiple, simultaneous interactions, and allows stochastic sampling of latent actions at inference time, capturing multimodality (e.g., moving left or right from the same state). The proposed module explicitly separates the modeling of latent actions (which encapsulate the stochastic aspects) from the dynamics prediction.

Formally, the CONTEXT module takes as input a sequence of particle sets across $T + 1$ frames, **optionally** conditioned on external signals $\{c_t\}_{t=0}^T$ (e.g., control actions, goal images, or language instructions). It outputs a sequence of per-particle latent contexts:

$$\mathcal{K}_\psi(\{[\{z_{\mathrm{fg}}^{m,t}\}_{m=0}^{M-1}, z_{\mathrm{bg}}^t, c_t]\}_{t=0}^T) = \{[\{z_{c,\mathrm{fg}}^{m,t}\}_{m=0}^{M-1}, z_{c,\mathrm{bg}}^t]\}_{t=0}^{T-1}.$$

The CONTEXT module is implemented as a *causal spatio-temporal transformer* ((Zhu et al., 2024), Appendix A.4.5), which jointly processes particles across space and time while ensuring autoregressive temporal dependencies. It is composed of two complementary heads: (1) **Latent inverse dynamics** $p_\psi^{\mathrm{inv}}(z_c^t \mid z^{t+1}, z^t, \ldots, z^0, c_t)$, which predicts the latent action responsible for the transition between consecutive states; (2) **Latent policy** $p_\psi^{\mathrm{policy}}(z_c^t \mid z^t, \ldots, z^0, c_t)$, which models the distribution of latent actions conditioned on the current state.

The latent policy serves as a prior that regularizes the inverse dynamics via a KL-divergence penalty in the VAE objective (Appendix A.4.6). Specifically, the latent actions are modeled as Gaussian distributions, $z_c \sim \mathcal{N}(\mu_c, \sigma_c^2)$, parameterized by the context module. At training time, latent actions are obtained through the inverse dynamics head, ensuring consistency with observed transitions. At inference time, latent actions can instead be sampled directly from the latent policy prior, enabling stochastic rollouts of the world model. Conditioning on external signals (global actions, language instructions or image-based goals) *within* the latent context module maps global scene-level signals into per-particle latent actions. For instance, given a language instruction, $\mathcal{K}_\psi$ learns to translate it into per-particle latent actions that drive the dynamics towards satisfying the instruction. When no external conditioning is provided, $\mathcal{K}_\psi$ simply infers latent actions from past particle trajectories. When conditioned on a goal image or a language instruction, sampling from the latent policy can be further utilized for planning in the particles space, as we demonstrate later in the experiments section. In Appendix A.4.3 we describe how the global action, language and image conditioning mechanisms are implemented. Finally, we note that the proposed novel CONTEXT module is broadly applicable to general-purpose, non-object-centric architectures utilizing patch-based representations, as demonstrated in the experiments section. The CONTEXT module is illustrated in Figure 3.

**DYNAMICS $\mathcal{F}_\xi$**: The dynamics module implements the VAE's autoregressive dynamics prior $p_\xi(z^t \mid z^{t-1}, \ldots, z^0)$. It predicts the particles at the next timestep conditioned on the current particles and their corresponding latent actions provided by the context module:

$$\mathcal{F}_\xi\Big(\big\{[\{z_{\text{fg}}^{m,t}\}_{m=0}^{M-1}, z_{\text{bg}}^t, z_c^t]\big\}_{t=0}^{T-1}\Big) = \big\{[\{\hat{z}_{\text{fg}}^{m,t}\}_{m=0}^{M-1}, \hat{z}_{\text{bg}}^t]\big\}_{t=1}^{T}.$$

It is implemented as a causal spatio-temporal transformer, where particles are conditioned on their corresponding latent actions via AdaLN (Zhu et al., 2024). The module outputs distribution parameters serving as the prior in the KL-divergence between the encoder posterior and dynamics prior.[1]

Unlike DDLP (Daniel & Tamar, 2024), LPWM retains all $M$ encoded particles with their identities (patch origins) across timesteps, removing the need to track particles over time. This results in an implicit regime where particles can move in a certain region around their origin, balancing between two extremes: patch-based methods (e.g., VideoGPT (Yan et al., 2021)), where particles are fixed patches with evolving features, and object-centric particle models (Daniel & Tamar, 2024), which track a subset of free-moving particles with explicit attributes that can traverse the entire canvas. We discuss tracking limitations and the implications of this regime in Appendix A.4.4.

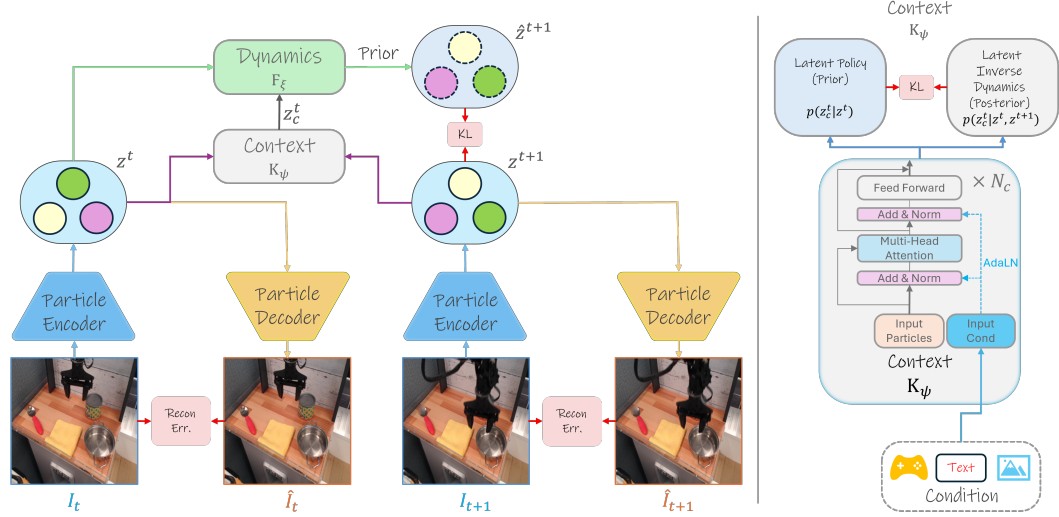

Figure 3: Latent Particle World Model architecture. Left: Input frames are encoded into particle sets by the ENCODER and decoded back to images by the DECODER. The CONTEXT module then processes the particles to sample latent actions, which are combined with the particles in the DYNAMICS module to predict next-step particle states. Right: The CONTEXT module models the per-particle latent action distribution. During training, we use the latent inverse dynamics head, while at inference, the latent policy is employed for sampling.

**Optimization and Training Details**: LPWM, following DDLP, is trained by maximizing a temporal ELBO, or minimizing the sum of reconstruction errors and KL-divergences, decomposed into a *static* term for the first frame and a *dynamic* term for subsequent frames: $\mathcal{L}_{\text{LPWM}} = -\sum_{t=0}^{T-1} ELBO(x_t = I_t) = \mathcal{L}_{\text{static}} + \mathcal{L}_{\text{dynamic}}$. The static term covers the single-frame setting, computing per-particle KL with respect to fixed priors, and adds regularization on particle transparency. The dynamic term includes KL losses for both latent actions and predicted future particles. All KL terms are evaluated in closed-form. Both losses also include frame-wise reconstruction loss: pixel-wise MSE for simulated datasets, or MSE and LPIPS (Hoshen et al., 2019) for real-world data. A key difference from DDLP is that KL contributions are masked using the particle transparency attribute (Lin et al., 2020a), so only visible particles affect the KL loss. Full loss details are provided in Appendix A.4.6. For all experiments, the dimension of the latent actions is set to $d_{\text{ctx}} = 7$. Models are optimized end-to-end with Adam (Kingma & Ba, 2014) with a learning rate of $8 \times 10^{-5}$ and implemented in PyTorch (Paszke et al., 2017). Hyperparameter details are in Appendix A.9. Code and pretrained models are available at `https://github.com/taldatech/lpwm`.

---

[1]The priors for the first timestep particles are fixed hyperparameters, consistent with DLP's single-image setup.

## 5 EXPERIMENTS

We design our experimental suite with the following key goals: (1) benchmark LPWM on unconditional and conditional video prediction across real-world and synthetic datasets; (2) analyze the impact of LPWM's design choices through ablation studies; and (3) demonstrate a practical imitation learning application on diverse, multi-object, long-horizon tasks and environments.

### 5.1 SELF-SUPERVISED OBJECT-CENTRIC VIDEO PREDICTION AND GENERATION

We evaluate LPWM across multiple video prediction settings, including unconditional, action-conditioned, and language-conditioned scenarios. Additional demonstrations of image conditioning and multi-view training, particularly for goal-conditioned imitation learning, are presented in Section 5.2.

**Datasets:** we evaluate our approach on a diverse set of datasets, spanning real-world and simulated domains with varying dynamics and interaction densities. Simulated datasets include `OBJ3D`, featuring dense interactions and deterministic 3D physics (Lin et al., 2020a), `PHYRE`, a sparse interaction 2D physical reasoning benchmark with deterministic dynamics (Bakhtin et al., 2019), and `Mario`, a stochastic 2D dataset with dense interactions from expert Super Mario Bros gameplay videos (Smirnov et al., 2021). Real-world datasets encompass robotic datasets such as `Sketchy` with sparse stochastic interactions (Cabi et al., 2019), `BAIR` and `Bridge` featuring dense, stochastic robotic manipulation with and without language instructions (Ebert et al., 2017a; Walke et al., 2023), and `LanguageTable` featuring language-guided, dense object rearrangements (Lynch et al., 2023). Unless stated otherwise, all datasets are trained at $128 \times 128$ resolution. We provide a detailed description of each dataset in Appendix A.7.

**Baselines:** Our main baseline is a non-object-centric patch-based dynamics VAE (DVAE) world model, where "particles" correspond to fixed grid patch embeddings matching the number of LPWM particles, $M$. This baseline shares the same architecture and parameter count as LPWM and supports identical conditioning but lacks explicit attribute modeling. It closely resembles large-scale video generation models using patch-based tokenization (Yan et al., 2021; Yang et al., 2024b). Unlike pre-trained or quantized patch embeddings, ours are learned end-to-end like LPWM's particles, with a higher latent dimension to offset the absence of object-centric structure. When applicable, we also compare against recent object-centric video prediction methods, including the slot-based PlaySlot (Villar-Corrales & Behnke, 2025) for latent-action-conditioned tasks; and for deterministic dynamics datasets, the patch-based G-SWM (Lin et al., 2020a), the slot-based SlotFormer/OCVP (Wu et al., 2022b; Villar-Corrales et al., 2023), and the particle-based DDLP (Daniel & Tamar, 2024). Extended baseline details are provided in Appendix A.8.

**Metrics:** For latent-action-conditioned video prediction and datasets with deterministic dynamics, we report standard visual similarity metrics—PSNR, SSIM (Wang et al., 2004), and LPIPS (Zhang et al., 2018)—to compare generated sequences against ground truth[2]. For stochastic video generation, we compute the Fréchet Video Distance (FVD, Unterthiner et al. (2018); Hu (2023)) to evaluate the distributional similarity between generated and real video sets.

**Results:** LPWM outperforms all baselines on LPIPS and FVD metrics across stochastic dynamic datasets under various conditioning settings (Table 2). It effectively preserves *object permanence* throughout generation (Figure 1) and models complex object interactions, unlike competing methods that exhibit blurring or deformation. LPWM also supports multi-modal sampling, producing diverse plausible rollouts from identical initial conditions (see Appendix A.10 and videos). Compared to the slot-based PlaySlot baseline, which suffers from object drifting and blurry reconstructions due to global latent actions and limited number of slots, LPWM's per-particle latent actions and low-dimensional representation scale effectively to many-object scenarios. DVAE, a non-object-centric baseline, performs well on synthetic data but lacks robustness on real-world datasets, highlighting the advantages of object-centric modeling. Finally, we demonstrate that a compact LPWM model trained on `BAIR-64` matches larger video generation models in FVD (89.4, Table 9), emphasizing how object-centric inductive biases enable superior modeling of object interactions beyond what

---

[2]Our evaluation follows the DDLP protocol (Daniel & Tamar, 2024) using the open-source PIQA library (Rozet, 2022) for perceptual metrics.

scale alone achieves. Extended results are in Appendix A.10 and videos are available: `https://taldatech.github.io/lpwm-web`.

| Dataset | Sketchy-U | | | | BAIR-U | | | | Mario-U | | | |
|---|---|---|---|---|---|---|---|---|---|---|---|---|
| Setting | $t:20, c:6, p:44$ | | | | $t:16, c:1, p:15$ | | | | $t:20, c:6, p:34$ | | | |
| | PSNR↑ | SSIM↑ | LPIPS↓ | FVD↓ | PSNR↑ | SSIM↑ | LPIPS↓ | FVD↓ | PSNR↑ | SSIM↑ | LPIPS↓ | FVD↓ |
| DVAE | 25.75±3.85 | 0.86±0.08 | 0.113±0.06 | 140.06 | 26±2.2 | 0.90±0.03 | 0.063±0.02 | 164.41 | 23.35±4.28 | 0.93±0.04 | 0.087±0.05 | 277.41 |
| PlaySlot | 22.63±3.90 | 0.80±0.09 | 0.275±0.06 | — | 17.56±1.50 | 0.57±0.05 | 0.483±0.03 | — | 16.38±2.78 | 0.68±0.1 | 0.314±0.09 | — |
| LPWM (Ours) | 28.41±3.8 | 0.91±0.06 | **0.070±0.04** | 85.45 | 25.66±1.52 | 0.89±0.02 | 0.062±0.02 | 163.91 | 27.50±5.52 | 0.95±0.04 | **0.035±0.02** | 195.95 |

| Dataset | Sketchy-A | | | LanguageTable-A | | | LanguageTable-L | Bridge-L |
|---|---|---|---|---|---|---|---|---|
| Setting | $t:20, c:6, p:44$ | | | $t:20, c:1, p:15$ | | | $t:20, c:1, p:15$ | $t:24, c:1, p:29$ |
| | PSNR↑ | SSIM↑ | LPIPS↓ | PSNR↑ | SSIM↑ | LPIPS↓ | FVD↓ | FVD↓ |
| DVAE | 25.33±4.2 | 0.85±0.09 | 0.111±0.06 | 29.29±5.28 | 0.94±0.04 | **0.038±0.02** | 26.78 | 146.85 |
| LPWM (Ours) | 27.06±4.26 | 0.88±0.09 | **0.083±0.05** | 29.5±5.06 | 0.94±0.04 | **0.037±0.02** | **15.96** | **47.78** |

Table 2: Quantitative results on latent-action-conditioned (U), action-conditioned (A), and language-conditioned (L) video prediction. FVD is reported for stochastic generation by sampling from the latent policy. $t$ is the training horizon, $c$ is the conditional frames at inference, and $p$ is the predicted frames at inference.

**Ablation Analysis:** We conduct a series of ablation studies on our design choices, including global versus per-particle latent actions, the dimensionality of the latent action vector, and types of positional embeddings. As detailed in Appendix A.10.3, our results demonstrate that per-particle latent actions are essential for achieving strong performance and that the model is robust to latent action dimension near the effective particle dimension ($6 + d_{\text{obj}}$). Furthermore, embedding positional information via AdaLN outperforms standard additive positional embeddings.

## 5.2 IMITATION LEARNING WITH PRE-TRAINED LPWM

Pre-training LPWM on actionless video datasets enables it to predict video dynamics using latent actions, suggesting that these latent actions capture actionable information. Assuming access to ground-truth actions and that the latent actions effectively encode dynamics, learning a simple mapping from latent actions to true actions may suffice to derive a policy, which we verify next. Formally, once paired video-action trajectories ($I_{0:T}, a_{0:T-1}$) are available (e.g., collected post-hoc), pre-trained LPWMs can be adapted to goal-conditioned imitation tasks using image-based goals (details in Appendix A.4.3). For each trajectory, image sequences are encoded by a frozen LPWM to produce per-particle latent actions $\{z_c^{m,t}\}_{m=0}^{M}$ via the latent inverse dynamics head $p_\psi^{\text{inv}}(z_c^t \mid z^{\leq T})$. Policy learning maps latent actions to global actions using a simple, compact, two-layer attention pooling transformer (Haramati et al., 2024). At inference, given a goal image and a rollout horizon $k$, LPWM is autoregressively unrolled for $k+1$ steps to generate particles and their $k$ latent actions. The trained mapping network outputs a sequence of $k$ global actions, which are executed sequentially in the environment (Zhao et al., 2023); this process is repeated until the maximum number of environment steps is reached. Implementation and training details are described in Appendix A.5.

| Task | EC Diffusion Policy | EC Diffuser | LPWM (Ours) | | Task | GCIVL | HIQL | LPWM (Ours) |
|---|---|---|---|---|---|---|---|---|
| 1 Cube | 88.7 ± 3 | 94.8 ± 1.5 | 92.7 ± 4.5 | | task1 | 84 ±4 | 80 ±6 | 100 ±0 |
| 2 Cubes | 38.8 ± 10.6 | 91.7 ± 3 | 74 ± 4 | | task2 | 24 ±8 | 81 ±7 | 6 ±9 |
| 3 Cubes | 66.8 ± 17 | 89.4 ± 2.5 | 62.1 ± 4.4 | | task3 | 16 ±8 | 61 ±11 | 89 ±9 |

Table 3: Imitation learning results (success rates) on `PandaPush` (left) and `OGBench-Scene` (right).

**Environments**: `PandaPush` challenges manipulation of up to three cubes observed from two camera views, while `OGBench-Scene` evaluates long-horizon planning involving diverse objects such as drawers and buttons. We train a single LPWM per environment that *encompasses all tasks*, whereas for `PandaPush`, the baselines train separate policies for each task, effectively giving them an advantage by optimizing individually for each task.

**Results**: Table 3 summarizes performance compared to the two best baselines in each (full results in Tables 12 and 13). On `PandaPush`, LPWM outperforms all baselines except EC Diffuser and matches its performance on the 1-cube task. We employ the multi-view LPWM variant here, modeling particle dynamics from multiple views simultaneously, highlighting the framework's flexibility (see Appendix A.4.5). On `OGBench`, despite the challenge of highly suboptimal, unstructured 'play' data hindering behavioral cloning, our method achieves strong results on tasks involving up to four atomic behaviors (`task1` and `task3`), outperforming all baselines on these. For `task4` and `task5`, all methods fail (with the exception of HIQL attaining 20% success rate on `task4`). Although we employ a relatively simple policy, LPWM demonstrates competitive performance, underscoring its potential for decision-making applications. Figure 4 visualizes an example imagined

trajectory alongside environment execution on `OGBench`, and rollout videos are available on the project website. Full results, baseline details and more visualizations are in Appendix A.10.4.

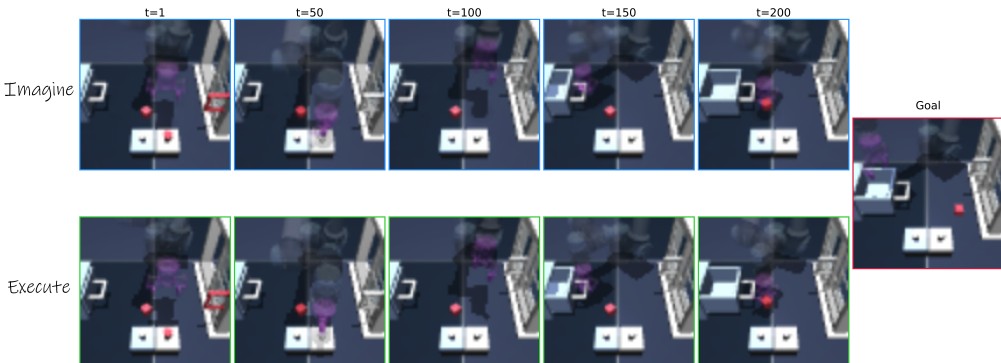

Figure 4: LPWM generated goal-conditioned imagined trajectories (top) and actual environment executions (bottom) through a learned mapping to actions on `OGBench-Scene`. The imagined trajectories closely match the actual executions, demonstrating LPWM's predictive accuracy.

## 6 CONCLUSION

We introduced Latent Particle World Model (LPWM), advancing self-supervised object-centric world modeling for real-world data. LPWM discovers keypoints, bounding boxes, and masks in a fully self-supervised fashion, decomposing scenes into latent particles whose temporal evolution is modeled by novel latent action and dynamics modules. This design enables state-of-the-art stochastic object-centric video generation while flexibly supporting action, language, and image conditioning, as well as multi-view inputs. LPWM shows strong potential for decision-making tasks, including imitation learning as demonstrated here.

**Limitations:** LPWM presently depends on datasets exhibiting small camera motion and recurring scenarios, such as robotics or video games; it is not yet applicable to general-purpose large-scale video data. Future work could address scaling to diverse datasets, unified multi-modal conditioning (e.g., simultaneous action, language, and image signals), and integration with explicit reward modeling for reinforcement learning.

## 7 ETHICS STATEMENT

This work introduces a video generation model evaluated on both simulated and real-world robotics datasets. As discussed in the limitations above, our model is not demonstrated on general-purpose video data or applied to sensitive content. All datasets used are either publicly available or collected in controlled, non-sensitive environments. We do not foresee ethical or societal risks arising from this work as presented; however, as with any generative model, future extensions to broader or less-controlled domains should carefully consider potential misuse and ensure responsible deployment.

## 8 REPRODUCIBILITY STATEMENT

We strive to facilitate reproducibility and transparency in self-supervised object-centric world modeling. To this end, we provide code excerpts throughout the appendix, along with extended implementation details and the full list of hyperparameters in Appendix A.9. Source code and pre-trained model checkpoints are available. These resources aim to make it straightforward for others to build upon our framework.

### ACKNOWLEDGMENTS

This work is supported by ONR MURI N00014-24-1-2748 and ERC grant Bayes-RL (101041250). Views and opinions expressed are those of the authors only and do not necessarily reflect those of the European Union or the European Research Council Executive Agency (ERCEA). Neither the European Union nor the granting authority can be held responsible for them.

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

# A APPENDIX

## A.1 LARGE LANGUAGE MODELS (LLMS) ASSISTANCE DISCLOSURE

We used large language models (LLMs) to assist in polishing the writing and improving grammar on a sentence level. All suggestions were reviewed and approved by the authors.

## A.2 PRELIMINARIES: SPATIAL SOFTMAX (SSM) AND SPATIAL TRANSFORMER NETWORK (STN)

We first review two core building blocks of the Deep Latent Particles (DLP, Daniel & Tamar, 2022a; 2024) model, an object-centric latent representation with disentangled attributes, before formally defining DLP. The *Spatial Softmax* (SSM, Jakab et al., 2018; Finn et al., 2016b) is commonly used for self-supervised extraction of keypoints from feature maps. The *Spatial Transformer Network* (STN, Jaderberg et al., 2015) provides a differentiable mechanism for spatial transformations: given a set of keypoint locations, it enables the model to extract localized patches from the image and to recompose the image from such patches using parameterized affine transformations.

**Spatial Softmax (SSM).** The spatial softmax, also known as the soft-argmax, can be viewed as a differentiable relaxation of the argmax operator: rather than selecting a single coordinate, it computes the expected coordinate under a probability distribution. Given a heatmap $\tilde{\mathcal{H}} \in \mathbb{R}^{H \times W}$, typically obtained from CNN feature maps of an image or patch, the softmax function is applied over the spatial dimensions to normalize $\tilde{\mathcal{H}}$ into a probability distribution $\mathcal{H}$. Each entry $h_{ij} = \mathcal{H}(i, j)$ then represents the probability of a keypoint being located at position $(i, j)$. From this distribution, the mean coordinate $(\mu_x, \mu_y)$ and the covariance values $\sigma_x^2, \sigma_y^2$, and $\sigma_{xy}$, following Sun et al. (2022), are computed:

$$\mu_x = \sum_i x_i \sum_j \mathcal{H}(i, j), \qquad \mu_y = \sum_j y_j \sum_i \mathcal{H}(i, j),$$

$$\sigma_x^2 = \sum_{ij} (x_i - \mu_x)^2 \mathcal{H}(i, j), \qquad \sigma_y^2 = \sum_{ij} (y_j - \mu_y)^2 \mathcal{H}(i, j),$$

$$\sigma_{xy} = \sum_{ij} (x_i - \mu_x)(y_j - \mu_y) \mathcal{H}(i, j).$$

Here, $\sum_j \mathcal{H}(i, j)$ and $\sum_i \mathcal{H}(i, j)$ correspond to the marginal distributions along each axis. The coordinate grids $\{x_i\}$ and $\{y_j\}$ are defined as normalized continuous values, typically spanning $[-1, 1]$ across the width and height, rather than raw pixel indices. The process is illustrated in Figure 5 and we provide a PyTorch-style code in Figure 6. Intuitively, sharply peaked activations yield low covariance values, typically corresponding to salient structures such as objects, corners, or edges. In contrast, broadly spread activations tend to produce high covariances, which are characteristic of background or less informative regions. Thus, covariance values provide a natural criterion for detecting and filtering meaningful locations. Since the SSM is fully differentiable, the heatmaps are optimized end-to-end through the reconstruction objective, encouraging the model to attend to the most informative regions of the scene.

**Spatial Transformer Network (STN).** A Spatial Transformer Network (STN; Jaderberg et al., 2015) is a learnable module that performs spatial transformations on input data in a fully differentiable manner. Such transformations include translation, scaling, rotation, and more general warping. In our context, we focus on the core differentiable operation underlying STNs: **grid sampling**.

Given an input image $I \in \mathbb{R}^{C \times H \times W}$, an affine transformation matrix

$$A = \begin{bmatrix} a_{11} & a_{12} & t_x \\ a_{21} & a_{22} & t_y \end{bmatrix}$$

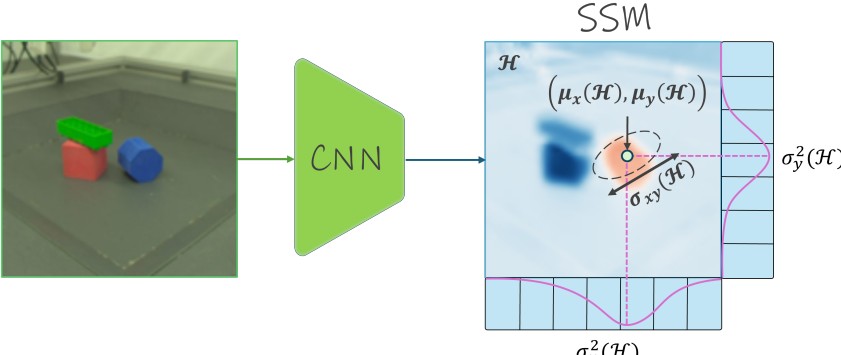

Figure 5: Spatial-softmax. Given a heatmap $\tilde{\mathcal{H}} \in \mathbb{R}^{H \times W}$, the softmax function is applied over the spatial dimensions to normalize $\tilde{\mathcal{H}}$ into a probability distribution $\mathcal{H}$. These values are then used to compute the expected coordinate values for each axis, and their covariance.

```python
def spatial_softmax(heatmap, kp_range=(-1, 1)):
    """
    Spatial Softmax with Marginalization for keypoint detection.
    Args:
        heatmap: [B, K, H, W] input heatmaps
        kp_range: coordinate range for normalization (default: (-1, 1))
    Returns:
        kp: [B, K, 2] expected keypoint coordinates [y, x]
        var: [B, K, 3] variance estimates [var_y, var_x, cov_yx]
    """
    batch_size, n_kp, height, width = heatmap.shape

    # 1. Flatten and apply softmax
    logits = heatmap.view(batch_size, n_kp, -1)  # [B, K, H*W]
    scores = torch.softmax(logits, dim=-1)
    scores = scores.view(batch_size, n_kp, height, width)  # [B, K, H, W]

    # 2. Create coordinate axes
    y_axis = torch.linspace(kp_range[0], kp_range[1], height,
                            device=scores.device).type_as(scores)
    x_axis = torch.linspace(kp_range[0], kp_range[1], width,
                            device=scores.device).type_as(scores)

    # 3. Marginalize over dimensions
    sm_h = scores.sum(dim=-1)  # [B, K, H] - marginalize over width
    sm_w = scores.sum(dim=-2)  # [B, K, W] - marginalize over height

    # 4. Compute expected coordinates
    kp_y = torch.sum(sm_h * y_axis, dim=-1)  # [B, K]
    kp_x = torch.sum(sm_w * x_axis, dim=-1)  # [B, K]
    kp = torch.stack([kp_y, kp_x], dim=-1)  # [B, K, 2]

    # 5. Compute variance: Var(X) = E[X^2] - (E[X])^2
    y_sq = (scores * (y_axis.unsqueeze(-1) ** 2)).sum(dim=(-2, -1))
    var_y = (y_sq - kp_y ** 2).clamp_min(1e-6)  # [B, K]

    x_sq = (scores * (x_axis.unsqueeze(-2) ** 2)).sum(dim=(-2, -1))
    var_x = (x_sq - kp_x ** 2).clamp_min(1e-6)  # [B, K]

    # 6. Compute covariance: Cov(X,Y) = E[XY] - E[X]E[Y]
    xy = (scores * (y_axis.unsqueeze(-1) * x_axis.unsqueeze(-2))).sum(dim=(-2, -1))
    cov_yx = xy - kp_y * kp_x  # [B, K]

    var = torch.stack([var_y, var_x, cov_yx], dim=-1)  # [B, K, 3]

    return kp, var
```

Figure 6: PyTorch-style code of Spatial Softmax for keypoint detection.

maps normalized target coordinates $(x^t, y^t) \in [-1, 1]^2$ in the output grid to source coordinates $(x^s, y^s)$ in the input:

$$\begin{bmatrix} x^s \\ y^s \end{bmatrix} = A \begin{bmatrix} x^t \\ y^t \\ 1 \end{bmatrix}.$$

The resulting sampling grid $\mathcal{G} = \{(x^s, y^s)\}$ specifies where to fetch pixels from the input image. The *grid sampling* operation then computes the transformed image $\hat{I}$ via bilinear interpolation:

$$\hat{I}(x^t, y^t) = \sum_{i,j} I(i, j) \max(0, 1 - |x^s - j|) \max(0, 1 - |y^s - i|).$$

This interpolation ensures that the transformation is differentiable with respect to both the sampling locations and the input image. In DLP, this mechanism is used in two ways:

1. **Encoding:** extracting glimpses from the input image, parameterized by each particle's attributes (position and scale).
2. **Decoding:** stitching back the reconstructed image from the decoded particle glimpses.

Thus, the encoder and decoder construct particle-specific affine transformations, generate the corresponding sampling grids, and warp the images or patches via differentiable grid sampling. Importantly, both grid generation and sampling are natively implemented in modern frameworks such as PyTorch (Paszke et al., 2017). A minimal PyTorch-style code snippet is provided in Figure 7.

```python
def spatial_transform(
    image, z_pos, z_scale, out_dims, inverse=False, eps=1e-9, padding_mode="zeros"
):
    """
    Differentiable spatial transform using grid sampling.

    Args:
        image: [B, C, H, W] input tensor.
        z_pos: [B, 2] position (tx, ty).
        z_scale: [B, 2] scale factors (sx, sy).
        out_dims: (B, C, H_out, W_out) desired output size.
        inverse: if False (default), encoding transform (image -> glimpse).
                 if True, decoding transform (glimpse -> image).
        eps: small constant for numerical stability.
        padding_mode: padding for out-of-bounds sampling.
    """
    # 1. Construct 2x3 affine transform matrix
    theta = torch.zeros(image.shape[0], 2, 3, device=image.device)

    # scaling
    theta[:, 0, 0] = z_scale[:, 1] if not inverse else 1 / (z_scale[:, 1] + eps)
    theta[:, 1, 1] = z_scale[:, 0] if not inverse else 1 / (z_scale[:, 0] + eps)

    # translation
    theta[:, 0, 2] = z_pos[:, 1] if not inverse else -z_pos[:, 1] / (z_scale[:, 1] + eps)
    theta[:, 1, 2] = z_pos[:, 0] if not inverse else -z_pos[:, 0] / (z_scale[:, 0] + eps)

    # 2. Construct grid and apply bilinear sampling
    grid = F.affine_grid(theta, torch.Size(out_dims), align_corners=False)
    return F.grid_sample(image, grid, mode='bilinear',
                         align_corners=False, padding_mode=padding_mode)
```

Figure 7: PyTorch-style code for differentiable warping with encoding (image→glimpse) and decoding (glimpse→image).

### A.3 DEEP LATENT PARTICLES (DLP)

We now formally define *deep latent particles* (DLP) (Daniel & Tamar, 2024) and their attributes. In the following sections, we describe how they are learned and used for dynamics modeling.

A foreground latent particle, illustrated in Figure 8, is defined as

$$z_{\text{fg}} = [z_p, z_s, z_d, z_t, z_f] \in \mathbb{R}^{6+d_{\text{obj}}},$$

where each component corresponds to a disentangled stochastic attribute: position $z_p$, scale $z_s$, depth $z_d$, transparency $z_t$, and visual features $z_f$. The background is represented by a single abstract particle $z_{\text{bg}}$, fixed at the center of the image and parameterized only by $d_{\text{bg}}$ background features. Formally,

$$z_{\text{bg}} \sim \mathcal{N}(\mu_{\text{bg}}, \sigma_{\text{bg}}^2) \in \mathbb{R}^{d_{\text{bg}}}.$$

We now detail the role of each attribute:

**Position** $z_p \in \mathbb{R}^2$: encodes the spatial location of the particle, i.e., its $(x, y)$ coordinates within $[-1, 1]$. Following object-centric models such as G-SWM (Lin et al., 2020a) and SCALOR (Jiang et al., 2019), $z_p$ is modeled as a Gaussian $\mathcal{N}(\mu_p, \sigma_p^2)$. In DLP, the prior for $z_p$ is derived from SSM over patches, which ensures that positions carry explicit spatial meaning.

**Scale** $z_s \in \mathbb{R}^2$: defines the particle's height and width (i.e., bounding box size). It is modeled as $\mathcal{N}(\mu_s, \sigma_s^2)$ and passed through a Sigmoid activation to constrain values to $[0, 1]$.

**Depth** $z_d \in \mathbb{R}$: specifies the relative ordering of particles when reconstructing the scene, and modeled as $\mathcal{N}(\mu_d, \sigma_d^2)$. While $z_d$ determines the compositing order of decoded objects, it does not necessarily correspond to physical 3D depth, since DLP is trained on monocular RGB inputs.

**Transparency** $z_t \in [0, 1]$: controls whether and to what extent a particle contributes to the reconstructed image. A value of $z_t = 0$ corresponds to a fully transparent (inactive) particle, $z_t = 1$ to a fully visible particle, and intermediate values capture partial visibility. Unlike many object-centric models that use a Bernoulli "presence" variable, we model transparency with a Beta distribution, $z_t \sim \text{Beta}(a_t, b_t)$. This has two key advantages: (1) the Beta distribution is continuous and reparameterizable, enabling stable gradient-based optimization without discrete relaxations, and (2) it naturally supports intermediate values, making it possible to represent partially occluded or semi-transparent objects. Moreover, like Gaussian and Bernoulli, the Beta distribution has a closed-form KL divergence that can be easily plugged in the VAE's objective function.

**Visual features** $z_f \in \mathbb{R}^{d_{\text{obj}}}$, $z_{\text{bg}} \in \mathbb{R}^{d_{\text{bg}}}$: encode the appearance of foreground particles, i.e., the keypoint's surrounding region, and the background, respectively. Both are modeled as Gaussian latents: $z_f \sim \mathcal{N}(\mu_f, \sigma_f^2)$ and $z_{\text{bg}} \sim \mathcal{N}(\mu_{\text{bg}}, \sigma_{\text{bg}}^2)$.

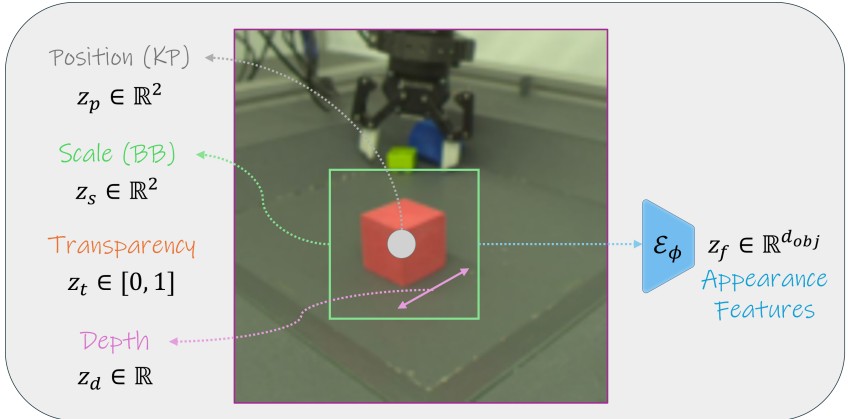

Figure 8: A Deep Latent Particle. Each component of a latent particle corresponds to a disentangled stochastic attribute: position $z_p$, scale $z_s$, depth $z_d$, transparency $z_t$, and visual features $z_f$. Further details are provided in Section 3 of the main text and Section A.3 of the Appendix.

## A.4 LATENT PARTICLE WORLD MODELS - EXTENDED METHOD DETAILS

In this section, we provide a detailed overview of our method.

Our goal is to design a *world model*, i.e., a dynamics model $\mathcal{F}(I_{0:T-1}, c) = \hat{I}_{T:T+\tau-1}$ that takes in a sequence of $T$ image observations $I_{0:T-1} \in \mathbb{R}^{T \times C \times H \times W}$ (a video), where $C$ is the number of image channels (typically 3 for RGB) and $H$ and $W$ are the height and width of the images respectively,

and *optionally* a sequence of conditioning signals $c$ (e.g., action sequence or language instruction), and predicts a sequence of future observations $\tau$ autoregressively $\hat{I}_{T:T+\tau-1} \in \mathbb{R}^{\tau \times C \times H \times W}$. We note that our world model need not be conditioned on $c$ and can be trained only with videos, i.e. $c = \emptyset$, as we explain later in the section. As the original image pixel space is high-dimensional, we propose an end-to-end latent world model, termed Latent Particle World Models (LPWM), which learns a compact self-supervised object-centric latent representation for the images, based on an improved version of the Deep Latent Particles (DLP, Daniel & Tamar, 2022a; 2024) representation, DLPv3, and a novel dynamics module learned over the latent particle space. The model is trained end-to-end such that the representation is trained to be predictable by the dynamics module, and as such does not require pre-trained image tokenization of any sort.

The **Latent Particle World Model (LPWM)** consists of four components, jointly trained end-to-end as a variational autoencoder (VAE, Kingma & Welling, 2014): the ENCODER $\mathcal{E}_\phi$, the DECODER $\mathcal{D}_\theta$, the CONTEXT $\mathcal{K}_\psi$ and the DYNAMICS $\mathcal{F}_\xi$. The general pipeline operates as follows: input frames are first encoded by the ENCODER into sets of particles, which are then decoded by the DECODER to reconstruct images and compute the reconstruction loss. The resulting sequence of latent particles is passed to the CONTEXT module, which generates distributions over latent actions. The sampled latent actions—together with the particles themselves—are processed by the DYNAMICS module to predict the next-step particle states, where the KL is computed per-particle. Below, we provide a high-level overview of each component and a detailed description is provided in subsequent sections. A high-level schematic of the encoding and decoding process is shown in Figure 9, and an overview of the full architecture is shown in Figure 3.

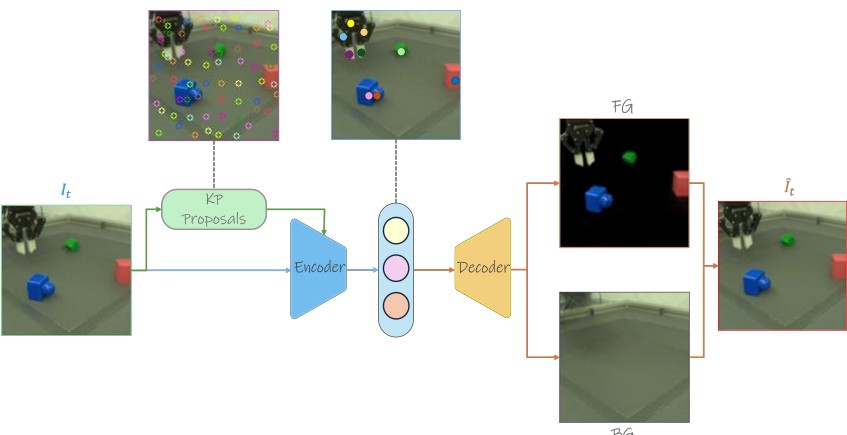

Figure 9: Encoding and decoding particles in DLP. Input image is first used to generate *keypoint proposals*, that, jointly with input image are used to encode a set of particles by the ENCODER, which is then decoded by the DECODER. Further details are provided in Section 4 of the main text and Section A.4 of the Appendix.

**ENCODER (Section A.4.1)**: corresponds to the VAE's approximate posterior $q_\phi(z|x)$. It takes as input an image frame and outputs a set of latent particles:

$$\mathcal{E}_\phi(x = I_t) = [\{z_{\text{fg}}^{m,t}\}_{m=0}^{M-1}, z_{\text{bg}}^t].$$

Each frame $I_t$ is represented by $M$ foreground latent particles $\{z_{\text{fg}}^{m,t}\}_{m=0}^{M-1}$, where each particle originates from per-patch learned keypoint (see Section A.4.1), and one background particle $z_{\text{bg}}^t$. Unlike DDLP, particle filtering to a subset $L \leq M$ is deferred to the decoder to preserve particle identities and avoid explicit tracking in downstream modules. Foreground particles are parameterized as

$$z_{\text{fg}}^m \in \mathbb{R}^{6+d_{\text{obj}}},$$

where the first six dimensions capture explicit attributes (e.g., spatial coordinates, scale, transparency), and the remaining $d_{\text{obj}}$ dimensions represent appearance features. The background particle is defined as

$$z_{\text{bg}} \in \mathbb{R}^{d_{\text{bg}}},$$

with $d_{\text{bg}}$ encoding the global appearance of the background.

**DECODER (Section A.4.2)**: corresponds to the VAE's likelihood $p_\theta(x|z)$. It takes as input a set of $L \leq M$ foreground particles together with a background particle, and reconstructs an image frame:

$$\mathcal{D}_\theta([\{z_{\text{fg}}^{l,t}\}_{l=0}^{L-1}, z_{\text{bg}}^t]) = \hat{I}_t.$$

Here, $L$ denotes the number of foreground particles provided to the decoder, which can be smaller than the $M$ particles produced by the encoder. Particles can be filtered based on their confidence – e.g., using their variance scores (Daniel & Tamar, 2024), or transparency values prior to rendering the frame, with the purpose of reducing the memory footprint without degrading the reconstruction performance.

**CONTEXT (Section A.4.3)**: a novel mechanism for modeling latent actions, i.e., the transitions that move a particle from $z_i^t$ to $z_i^{t+1}$. These latent actions provide the dynamics model with per-particle context, capturing events such as the stochastic movement of a gripper in a robotic video or the appearance of a new object in the scene. Throughout the paper, we use the terms *latent context* and *latent actions* interchangeably. Formally, the module takes as input a sequence of particle sets across $T + 1$ frames, optionally conditioned on external signals $\{c_t\}_{t=0}^T$ (e.g., control actions or language instructions), and outputs a sequence of per-particle latent contexts:

$$\mathcal{K}_\psi(\{[\{z_{\text{fg}}^{m,t}\}_{m=0}^{M-1}, z_{\text{bg}}^t, c_t]\}_{t=0}^T) = \{[\{z_{c,\text{fg}}^{m,t}\}_{m=0}^{M-1}, z_{c,\text{bg}}^t]\}_{t=0}^{T-1}.$$

The CONTEXT module consists of two complementary heads:

- **Latent inverse dynamics** $p_\psi^{\text{inv}}(z_c^t \mid z^{t+1}, z^t, \ldots, z^0, c_t)$, which predicts the latent action responsible for the transition between consecutive states.

- **Latent policy** $p_\psi^{\text{policy}}(z_c^t \mid z^t, \ldots, z^0, c_t)$, which models the distribution of latent actions given the current state.

In practice, the model does not output particles directly, but instead produces the parameters of their predictive distribution (e.g., Gaussian means and variances). The latent policy acts as a prior that regularizes the inverse dynamics via a KL-divergence term in the VAE objective (Section A.4.6). During training, latent actions are inferred through the inverse dynamics head. At inference time, actions can instead be sampled from the policy prior, enabling stochastic rollout of the world model.

**DYNAMICS (Section A.4.4)**: corresponds to the VAE's autoregressive dynamics prior $p_\xi(z^t|z^{t-1}, \ldots, z^0)$. It predicts the next-step particles conditioned on the current particles and their associated latent actions:

$$\mathcal{F}_\xi(\{[\{z_{\text{fg}}^{m,t}\}_{m=0}^{M-1}, z_{\text{bg}}^t, z_c^t]\}_{t=0}^{T-1}) = \{[\{\hat{z}_{\text{fg}}^{m,t}\}_{m=0}^{M-1}, \hat{z}_{\text{bg}}^t]\}_{t=1}^T.$$

Similarly to the context module, the model outputs the parameters of the distributions, which serve as the prior in the KL-divergence calculation part of the VAE training objective.

**LOSS (Section A.4.6)**: Latent Particle World Models are trained by maximizing a temporal ELBO, which decomposes into a *static* term (for the first frame) and a *dynamic* term (for subsequent frames). For brevity, we omit the particle index $m$, and note that both dynamics and context losses are summed over all $M$ particles.

**Static ELBO.** For the initial frame $x_0$, we optimize

$$\mathcal{L}_{\text{static}} = \mathcal{L}_{\text{rec}}(x_0, \hat{x}_0) + \beta_{\text{KL}} \, \text{KL}\big(q_\phi(z^0 \mid x_0) \, \| \, p(z^0)\big) + \beta_{\text{reg}} \, \mathcal{L}_{\text{reg}}(z_t^0),$$

where $z^0$ denotes the set of particle attributes and features. The KL is computed in a *masked form*, where each particle's contribution is weighted by its transparency attribute $z_t^m$, such that particles with $z_t^m \approx 0$ (inactive) have negligible effect. The transparency regularizer is defined as

$$\mathcal{L}_{\text{reg}} = \sum_{m=0}^{M-1} (z_t^m)^2,$$

which penalizes the total transparency values across particles and thus encourages sparse explanations of the scene, i.e., only a small subset of particles remain active.

**Dynamic ELBO.** For frames $t \geq 1$, the loss is

$$
\begin{aligned}
\mathcal{L}_{\mathrm{dynamic}} = \sum_{t=1}^{T-1} \Big[ & \mathcal{L}_{\mathrm{rec}}(x_t, \hat{x}_t) \\
& + \beta_{\mathrm{dyn}} \, \mathrm{KL}\Big( q_\phi(z^t \mid x_t) \,\|\, p_\xi\big(z^t \mid z^{<t}, z_c^{<t}\big)\Big) \\
& + \beta_{\mathrm{ctx}} \, \mathrm{KL}\Big( p_\psi^{\mathrm{inv}}(z_c^t \mid z^{\leq t+1}) \,\|\, p_\psi^{\mathrm{policy}}(z_c^t \mid z^{\leq t})\Big)\Big].
\end{aligned}
\tag{1}
$$

where $z_c^t$ denotes latent context (action) variables. The dynamics KL is masked as above, while the context KL is not, allowing context variables to also explain transitions between active and inactive states.

**Priors.** The static prior parameters are fixed: Gaussian means and covariances for attributes and visual features, and $(a, b)$ parameters of a Beta distribution for transparency (see Appendix A.9).

**Reconstruction.** The reconstruction term is defined as pixel-wise MSE in simulated environments, and as a perceptual loss in real-world data:

$$
\mathcal{L}_{\mathrm{rec}} = \begin{cases} \|x - \hat{x}\|_2^2, & \text{for simulated environments,} \\ \|x - \hat{x}\|_2^2 + \gamma \, \|\phi(x) - \phi(\hat{x})\|_2^2, & \text{for real-world datasets,} \end{cases}
$$

where $\phi(\cdot)$ denotes VGG features as in LPIPS (Hoshen et al., 2019), and $\gamma = 0.1$ controls the perceptual loss contribution.

In the following sections, we describe the technical and implementation details of each component, with the differences from DLPv2 and DDLP (Daniel & Tamar, 2024) highlighted in *italics*. In Section A.10.1, we compare the proposed DLPv3 to DLPv2 and DLP on image reconstruction in the single-image setting, demonstrating the effect of these modifications.

### A.4.1   ENCODER $\mathcal{E}_\phi$

We now describe the image encoding process from pixels to latent particles. The encoder operates per-frame (i.e., non-temporally; all frames are processed independently in parallel) and serves as the posterior of the VAE. The scheme largely follows DLPv2 (Daniel & Tamar, 2024); for completeness, we provide the details here and highlight the modifications introduced in DLPv3.

DLP learns an object-centric particle representation by disentangling *position* from *appearance* within a conditional VAE framework (Sohn et al., 2015). Specifically, keypoint proposals are first generated to represent candidate particle positions, after which additional attributes such as scale and transparency are extracted from regions centered around these proposals. The overall encoding steps are illustrated in Figure 10.

The encoder's role is to produce the posterior distribution over latent particles for a given image. Formally, it models the approximate posterior as

$$
q_\phi(z|x) = q_\phi(z_a|x) \times q_\phi(z_o, z_s, z_d, z_t|x, z_a) \times q_\phi(z_f|x, z_p, z_s),
$$

where $z_a$ denotes the keypoint proposals and $z_o$ the offsets that together form the particle positions $z_p = z_a + z_o$. This modular factorization improves performance: keypoint proposals from the spatial softmax (SSM) layer tend to capture regions of interest but are not guaranteed to align with object centers. Offsets, in contrast, are predicted by a neural network and can accurately adjust proposals to object centers–a property that is crucial for modeling object dynamics. The encoding process is hierarchical and involves of 3 steps: (1) *keypoint proposals* - a patch-based network with a spatial softmax layer generates candidate particle locations; (2) *attribute encoding* - offsets, scales, depths, and transparencies are inferred for each particle, and (3) *appearance encoding* - foreground particle features and global background features are extracted. We now describe each step in detail.

**Keypoint proposals.**   Given an input image $x \in \mathbb{R}^{C \times H \times W}$, we divide it into $M$ non-overlapping patches of size $D \times D$ (typically $D \in \{8, 16\}$). Each patch is processed by the *proposal encoder* $q_\phi(z_a|x)$, a lightweight convolutional neural network (CNN), followed by a spatial softmax (SSM)

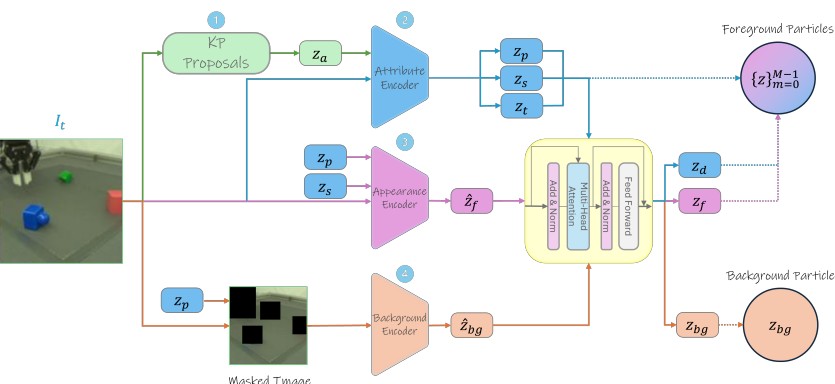

Figure 10: Encoding particles in DLP. The encoding process involves 4 steps: (1) *keypoint proposals* - a patch-based network with a spatial softmax layer generates candidate particle locations; (2) *attribute encoding* - offsets, scales, depths, and transparencies are inferred for each particle; (3) *appearance encoding* - foreground particle features, and (4) *background encoding* - global background features are extracted from the keypoint-masked image.

layer, which produces a single keypoint proposal $z_a$ per patch. *In DLPv2, these proposals were filtered down to a smaller set of size $L$ using the variance of the SSM distribution. In DLPv3, we instead postpone filtering until after other particle attributes have been estimated, which yields more reliable selection. In LPWM, filtering is deferred even further: particles are* never *removed in the encoder, but instead filtered in the decoder, ensuring that positional identity is preserved for downstream dynamics modeling. Nevertheless, for the single-image training setting, where only object-centric decomposition is learned and no dynamics are modeled, we retain the option of applying encoder-side filtering.*

**Attribute encoding.** To infer the position offset $z_o$, scale $z_s$, depth $z_d$, and transparency $z_t$ of each particle, we extract glimpses of size $S \times S$, where $S \geq D$, centered at the keypoint proposals $z_a$ using an STN.[3] These glimpses are processed by the *attribute encoder* $q_\phi(z_o, z_s, z_d, z_t | x, z_a)$, implemented as a small CNN followed by a fully connected layer, which outputs the distribution parameters described in Section A.3 for each particle. The encoder's weights are shared across all particles.

**Appearance encoding.** The visual features of each particle are extracted with the *appearance encoder* $q_\phi(z_f | x, z_p, z_s)$. As in the attribute stage, an STN is used to obtain glimpses of size $S$, but here the transformation is conditioned on both the particle position $z_p = z_a + z_o$ and the learned scale $z_s$, rather than a fixed patch ratio. This allows the glimpse size to adapt when objects are smaller or larger than the nominal $S \times S$ region. Since all stages rely on STN, the entire pipeline remains fully differentiable. The appearance encoder has the same architecture as the attribute encoder and outputs the Gaussian distribution parameters for the particle's visual features. In addition to object particles, we allocate a background particle anchored at the image center. Its features $z_{bg}$ are inferred by a dedicated *background encoder* $q_\phi(z_{bg} | x, z_p)$ which operates on a masked version of the input image. Specifically, the posterior keypoints $z_p$, along with their transparencies $z_t$, are used to generate $M$ masks of size $S \times S$, each masking out the region corresponding to a particle, leaving the background regions visible for encoding.

**DLPv3 encoding modifications.** *In DLPv3, we introduce several changes to the encoding process aimed at improving stability and performance:*

---

[3]The affine transformation in this stage uses a fixed scale corresponding to the patch size $S$, a predefined hyperparameter (typically 0.125 or 0.25 of the image size; we assume square images with $H = W$).

1. **Depth via particle attention.** Instead of predicting the depth attribute $z_d$ during the attribute encoding stage, we introduce an attention layer applied *after* attribute encoding. This attention layer takes input all particles, including the background particle, and outputs the depth values $\{z_d^m\}_{m=0}^{M-1}$. The motivation is that relative depth is inherently a global property, best estimated by jointly considering all particle positions and features rather than independently.

2. **Residual appearance encoding.** The same attention layer is also used to refine appearance features. Specifically, in the appearance encoding stage, we first compute a deterministic feature embedding $\hat{z}_f$ for each particle. The attention layer then outputs a residual $\Delta z_f$ and variance $\sigma_f^2$, such that the final appearance distribution is

$$q_\phi(z_f \mid x, z_p, z_s) = \mathcal{N}\big(z_f \mid \hat{z}_f + \Delta z_f, \ \sigma_f^2\big).$$

This residual modeling improves performance by allowing the network to adjust features based on contextual information from all particles.

3. **Stable transparency parameterization.** In DLPv2, the Beta distribution parameters $(a, b)$ for transparency were modeled as $a = \exp(y_a)$, $b = \exp(y_b)$, where $y_a, y_b$ are the outpus of the network, which could lead to excessively large concentration values and unstable training. In DLPv3, we reparameterize them as

$$a = r_{\max}\, \sigma(y_a) + r_{\min}\big(1 - \sigma(y_a)\big), \quad b = r_{\max}\, \sigma(y_b) + r_{\min}\big(1 - \sigma(y_b)\big),$$

where $\sigma(\cdot)$ is the sigmoid function, $r_{\min} = 10^{-4}$, and $r_{\max} = 100$. This constrains $(a, b)$ to a bounded range, leading to smoother and more stable optimization.

### A.4.2 DECODER $\mathcal{D}_\theta$

The decoder architecture is designed to mirror the object-centric structure of the latent representation. Each particle is decoded into a localized appearance patch, positioned and scaled according to its spatial attributes, while transparency and depth resolve visibility and occlusions. This compositional design parallels classical graphics pipelines, local rendering, spatial transformation, and alpha compositing, but is learned end-to-end from data, enabling the model to reconstruct complex scenes in a structured and interpretable manner.

The decoder models the likelihood

$$p_\theta(x \mid z) = p_\theta(x \mid z_{\text{fg}} = \{z_p, z_s, z_d, z_t, z_f\}, z_{\text{bg}})$$

and is composed of a *particle decoder* and a *background decoder*, as illustrated in Figure 11.

**Particle decoder.** Each particle is decoded independently into an RGBA (RGB + Alpha) glimpse $\tilde{x}_i^p \in \mathbb{R}^{S \times S \times 4}$, representing the reconstructed appearance of particle $i$. The particle decoder consists of a fully connected layer followed by a small upsampling CNN that maps the latent feature vector $z_f^{(i)}$ into this glimpse.

The alpha channel encodes a soft segmentation mask for the particle. The depth $z_d$ and transparency $z_t$ attributes modulate this mask, determining both the effective visibility and the compositing order of the particle. The spatial attributes $(z_p, z_s)$ specify the particle's position and scale, and are applied to the decoded glimpse using a Spatial Transformer Network (STN) to place it into the full-resolution canvas $\hat{x}_{\text{fg}}$.

The transparency and depth factorization process, which governs the stitching of multiple particles, is summarized in Figure 12.

**Background decoder.** The background is decoded from $z_{\text{bg}}$ using a standard VAE-style network: a fully connected layer followed by an upsampling CNN produces $\hat{x}_{\text{bg}}$, and the final reconstructed image is produced according to

$$\hat{x} = \alpha \odot \hat{x}_{\text{fg}} + (1 - \alpha) \odot \hat{x}_{\text{bg}},$$

where $\alpha$ is the effective mask obtained from the compositing process (Figure 12).

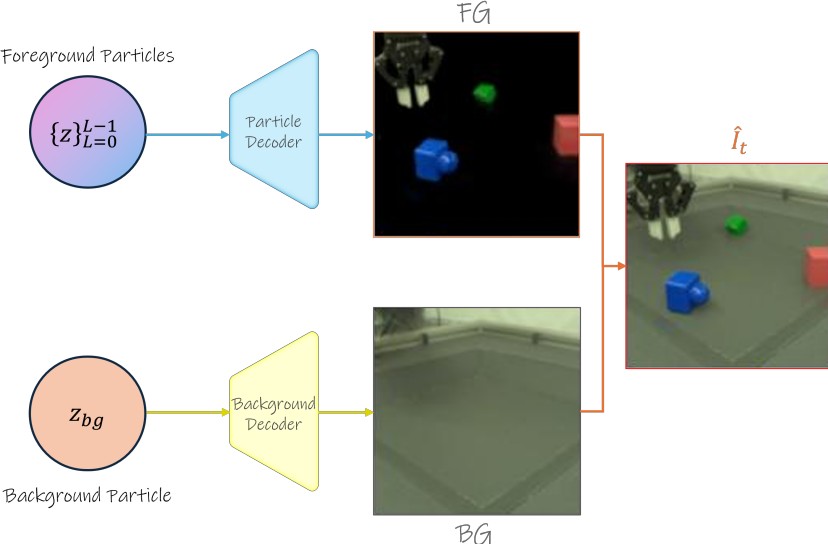

Figure 11: Decoding particles in DLP. Each particle is decoded into a localized appearance patch, positioned and scaled according to its spatial attributes, while transparency and depth resolve visibility and occlusions. The background is decoded with a standard upsampling CNN-based network. Finally, the foreground and background components are stitched using the effective alpha masks.

```python
def factor_alpha_map(alpha_obj, rgb_obj, z_t, z_d):
    # alpha_obj: [B, N, 1, h, w], per-particle alpha maps
    # rgb_obj:   [B, N, 3, h, w], per-particle RGB patches
    # z_t:       [B, N, 1], transparency attributes
    # z_d:       [B, N, 1], depth attributes

    # Apply transparency
    alpha_obj = alpha_obj * z_t

    # Mask RGB channels with alpha
    rgba_obj = alpha_obj * rgb_obj

    # Depth-based importance map
    importance = alpha_obj * sigmoid(-z_d)

    # Normalize importance weights
    importance = importance / (importance.sum(dim=1, keepdim=True) + 1e-5)

    # Composite objects according to importance
    objects_canvas = (rgba_obj * importance).sum(dim=1)

    # Background mask
    bg_mask = 1 - (alpha_obj * importance).sum(dim=1)

    return objects_canvas, bg_mask
```

Figure 12: PyTorch-style pseudocode of the transparency and depth factorization used for compositing particles.

*In DLPv3, we defer the particle filtering process to the decoder stage. Instead of rendering all $M$ particles, we only render a subset of $L \leq M$ particles, where filtering is based on the particles' positional variance and transparency. This design serves two purposes: (i) preserving the full set of particles after encoding, which is important for downstream dynamics modeling, and (ii) reducing computational and memory cost during rendering, since many particles may be inactive or redundant. The filtering procedure is described below.*

**Particle filtering.** *During training, we retain the top-$L$ particles with the lowest positional variance (i.e., highest spatial confidence). Formally, the positional variance of a particle is defined as*

$$V(z) = \sigma_x^2 + \sigma_y^2 + \sigma_{xy} + \sum_{j \in \{0,1\}} \log \sigma_{o,j}^2,$$

*where $\sigma_x^2, \sigma_y^2$, and $\sigma_{xy}$ denote the variance and covariance terms from the spatial softmax proposal, and $\sigma_{o,j}^2$ is the variance of the offset distribution predicted by the attribute encoder along axis $j$. We use the* log-variance *for the offset term to account for scale differences between the empirical variance from the spatial softmax and the learned offset uncertainty. This choice is motivated by prior work (Daniel & Tamar, 2022a), which demonstrated that particles with low positional variance tend to correspond to salient and meaningful parts of the scene, such as objects or object parts. At inference time, where particles are generated autoregressively, we simply discard all particles with zero transparency, i.e., $z_t = 0$.*

### A.4.3 CONTEXT $\mathcal{K}_\psi$

We now present the main novel additional component to the DLP framework—the CONTEXT module $\mathcal{K}_\psi$—designed to address the problem of *stochastic dynamics* in actionless videos. In such videos, scene dynamics are not fully determined by the first frames (e.g., a ball starting to roll in OBJ3D (Lin et al., 2020a) or CLEVRER (Yi et al., 2019)), but also by external signals such as actions (e.g., a robotic gripper moving in the BAIR dataset (Ebert et al., 2017a)).

A common approach to capture such stochastic transitions is to introduce *latent actions* (Menapace et al., 2021; Bruce et al., 2024; Gao et al., 2025; Villar-Corrales & Behnke, 2025). Typically, a latent action $z_c$ is learned in an autoencoding scheme: an inverse model infers $z_c^t = \mathcal{K}_\psi^{\text{inv}}(I_{t+1}, I_t)$ from two consecutive frames, and a decoder reconstructs the future frame $\hat{I}_{t+1} = \mathcal{D}_\theta(I_t, z_c^t)$, with training driven by reconstruction loss. To avoid degenerate solutions where $z_c^t$ memorizes $I_{t+1}$, $z_c$ is strongly regularized, either via a vector-quantization (VQ) bottleneck (Bruce et al., 2024; Ye et al., 2025) or a variational bottleneck with KL-regularization to a fixed prior (Gao et al., 2025). Crucially, in these designs, the latent action is *global*: a single vector encodes all changes between two frames. While this aligns with how agents are typically controlled (e.g., joint positions in robotics, discrete actions in video games), it is limited in multi-entity settings. For example, in Mario, enemies move independently of the player's true action space, and in robotics, contact events can induce secondary object interactions. A global action vector cannot naturally disentangle these local dynamics.

In this work, we introduce the CONTEXT module $\mathcal{K}_\psi$, a novel per-particle mechanism for latent action modeling. Unlike prior work (Villar-Corrales & Behnke, 2025; Gao et al., 2025), we model a latent action for each particle, directly governing the transition from $z_i^{m,t}$ to $z_i^{m,t+1}$. Regularization is not imposed via a fixed prior, but instead learned through a *latent policy*, which models the distribution of latent actions conditioned on the current state. This per-particle formulation enables the representation of multiple, simultaneous interactions, and allows stochastic sampling of latent actions at inference time, capturing multimodality (e.g., moving left or right from the same state).

Finally, we extend $\mathcal{K}_\psi$ to support external conditioning signals such as global actions (e.g., ground-truth gripper controls), language instructions, or image-based goals. Importantly, conditioning *within* the latent context module maps global scene-level signals into per-particle latent actions. For instance, given a language instruction, $\mathcal{K}_\psi$ learns to translate it into per-particle latent actions that drive the dynamics towards satisfying the instruction. When no external conditioning is provided, $\mathcal{K}_\psi$ simply infers latent actions from past particle trajectories.

Formally, the CONTEXT module takes as input a sequence of particle sets across $T + 1$ frames, **optionally** conditioned on external signals $\{c_t\}_{t=0}^T$ (e.g., control actions, goal images, or language instructions). It outputs a sequence of per-particle latent contexts:

$$\mathcal{K}_\psi(\{[\{z_{\text{fg}}^{m,t}\}_{m=0}^{M-1}, z_{\text{bg}}^t, c_t]\}_{t=0}^T) = \{[\{z_{c,\text{fg}}^{m,t}\}_{m=0}^{M-1}, z_{c,\text{bg}}^t]\}_{t=0}^{T-1}.$$

The CONTEXT module is implemented as a *causal spatio-temporal transformer* (Zhu et al., 2024), which jointly processes particles across space and time while ensuring autoregressive temporal dependencies. It is composed of two complementary heads:

- **Latent inverse dynamics** $p_\psi^{\text{inv}}(z_c^t \mid z^{t+1}, z^t, \ldots, z^0, c_t)$, which predicts the latent action responsible for the transition between consecutive states.

- **Latent policy** $p_\psi^{\text{policy}}(z_c^t \mid z^t, \ldots, z^0, c_t)$, which models the distribution of latent actions conditioned on the current state.

The latent policy serves as a prior that regularizes the inverse dynamics via a KL-divergence penalty in the VAE objective (Section A.4.6). Specifically, the latent actions are modeled as Gaussian distributions, $z_c \sim \mathcal{N}(\mu_c, \sigma_c^2)$, parameterized by the context module. At training time, latent actions are obtained through the inverse dynamics head, ensuring consistency with observed transitions. At inference time, latent actions can instead be sampled directly from the latent policy prior, enabling stochastic rollouts of the world model. When conditioned on a goal image or a language instruction, sampling from the latent policy can be further utilized for planning in the particles space, as we demonstrate in the experiments section (Section 5.2). The CONTEXT module is illustrated in Figure 3.

We now describe how different optional conditioning mechanisms are implemented.

**Action conditioning.** Given a sequence of $T$ global actions $\{c_t\}_{t=0}^{T-1} = \{a^t\}_{t=0}^{T-1}$ (e.g., robotic gripper joint positions), each action is projected to the transformer's inner dimension and then *repeated for all $M$ input particles*, such that the same global action conditions every particle at timestep $t$. Conditioning is applied via adaptive layer normalization (AdaLN, (Peebles & Xie, 2023)), enabling global actions to modulate the particle representations consistently across the scene.

**Language conditioning.** Given a language instruction, we embed its $K$ tokens with a pretrained T5-large model (Raffel et al., 2020) to obtain a sequence of embeddings $c_t = \{l_k\}_{k=0}^{K-1}$. These embeddings are projected to the transformer's inner dimension and appended to the $M$ particle embeddings along the sequence dimension, resulting in $M + K$ inputs at every timestep $t = 0, \ldots, T-1$. The joint set of particle and language embeddings is processed by self-attention, after which the language embeddings are discarded. We found this self-attention conditioning more effective than cross-attention, consistent with prior work in video generation (Yang et al., 2024b).

**Goal image conditioning.** Given a goal image $I_g$, we encode it into $M$ particles (plus a background particle) using the particle encoder ENCODER. The resulting goal particle set $c_t = \{z_g^m\}_{m=0}^{M-1}$ is repeated across all $T$ timesteps and used to condition the corresponding particles $\{z^i\}$ through AdaLN. This allows each particle to be guided toward its goal state in a temporally consistent manner.

### A.4.4 DYNAMICS $\mathcal{F}_\xi$

The dynamics module implements the VAE's autoregressive dynamics prior $p_\xi(z^t \mid z^{t-1}, \ldots, z^0)$. It predicts the particles at the next timestep conditioned on the current particles and their corresponding latent actions provided by the context module:

$$\mathcal{F}_\xi\left(\left\{[\{z_{\text{fg}}^{m,t}\}_{m=0}^{M-1}, z_{\text{bg}}^t, z_c^t]\right\}_{t=0}^{T-1}\right) = \left\{[\{\hat{z}_{\text{fg}}^{m,t}\}_{m=0}^{M-1}, \hat{z}_{\text{bg}}^t]\right\}_{t=1}^{T}.$$

Here $z_c^t$ denotes the latent actions at timestep $t$. The dynamics module $\mathcal{F}_\xi$ is implemented as a causal spatio-temporal transformer, where particles are conditioned on their corresponding latent actions through adaptive layer normalization (AdaLN (Zhu et al., 2024)).

As in the other components, $\mathcal{F}_\xi$ outputs distribution parameters that serve as the prior in the KL-divergence term between the posterior encoder $\mathcal{E}_\phi$ and the dynamics prior.[4]

*Differently from DDLP (Daniel & Tamar, 2024), LPWM does not rely on tracking a subset of particles across timesteps. Instead, it keeps the entire set of $M$ encoded particles along with their identities (i.e., the patches they originated from). This induces a particle-grid regime: each particle is constrained to move only within a local region around its original patch center, and when it reaches the limits of this region, its features are transferred to nearby particles. This mechanism is illustrated in Figure 13.*

---

[4]The priors for the particles in the first timestep are fixed hyperparameters, consistent with the single-image training setup of DLP.

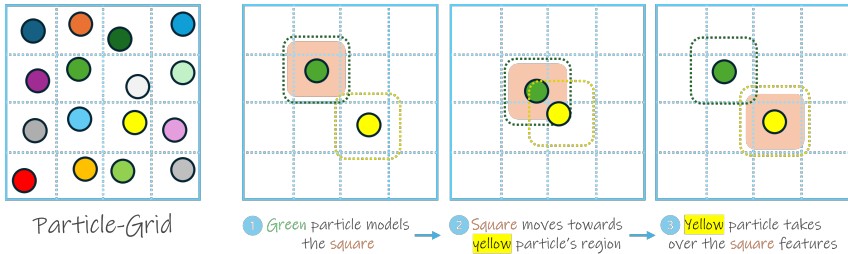

Figure 13: Particle-grid regime. Each particle is constrained to move only within a local region around its original patch center, and when it reaches the limits of this region, its features are transferred to nearby particles.

*This design balances between two extremes. On one side are patch-based approaches (e.g., VideoGPT (Yan et al., 2021)), where "particles" are fixed at patch centers and only patch features evolve over time. On the other side are object-centric particle models (Daniel & Tamar, 2024), where a subset of free-moving particles with explicit attributes (e.g., position) can traverse the entire canvas, but their identities must be tracked across time. The latter assumption may hold in controlled settings—for instance, videos with deterministic dynamics and moderate frame rates—but it fails in more general video data where actions or stochastic events occur.*

*As discussed in Daniel & Tamar (2024), relying on tracking introduces two key limitations: (1) the tracking algorithm assumes sufficiently small frame-to-frame displacements, which constrains the model to certain frame rates; and (2) since the tracked subset of particles is fixed, the model cannot naturally represent events such as new objects entering the scene without additional mechanisms (Lin et al., 2020a).*

*In contrast, purely patch-based dynamics models avoid these issues by predicting only fixed patch features without explicit attributes (e.g., keypoints). While more general, such models struggle to capture fine-grained object interactions and relations (Lin et al., 2020a; Wu et al., 2022b; Daniel & Tamar, 2024). LPWM, through its particle-grid design, aims to combine the generality of patch-based models with the expressivity of object-centric particles.*

### A.4.5 TRANSFORMER ARCHITECTURE AND MULTI-VIEW

**Spatio-temporal transformer.** Given a temporal input sequence of particle sets with shape $[B, T, M, D]$, where $B$ is batch size, $T$ is temporal horizon, $M$ is the number of particles, and $D$ is the embedding dimension, a standard transformer block applies multi-head attention over all $T \times M$ tokens, resulting in quadratic computation cost. To reduce this, LPWM employs a memory-efficient spatio-temporal attention mechanism (Ma et al., 2025; Zhu et al., 2024) (see Figure 14), which decomposes each spatio-temporal block into two stages: (1) a spatial block that processes all $M$ particles at each timestep independently ($[B \times T, M, D]$), and (2) a temporal block that captures temporal dependencies for each particle across time ($[B \times M, T, D]$).

For conditioning, we use adaptive layer normalization (AdaLN) (Peebles & Xie, 2023). Given a condition vector $c$ and intermediate feature $z$, AdaLN modulates transformer block activations as:

$$\alpha_1, \alpha_2, \beta_1, \beta_2, \gamma_1, \gamma_2 = \text{MLP}(c),$$

$$z = z + \alpha_1 \cdot \text{Self-Attention}(\gamma_1 \cdot \text{RMSNorm}(z) + \beta_1),$$

$$z = z + \alpha_2 \cdot \text{MLP}(\gamma_2 \cdot \text{RMSNorm}(z) + \beta_2).$$

This mechanism is used to incorporate positional and temporal information within the transformer in addition to other conditional inputs such as actions, language tokens or images. During training, we use teacher forcing (Williams & Zipser, 1989), while inference is performed autoregressively.

**Multi-view support.** We extend LPWM to jointly train on multiple camera views by *synchronizing* particle dynamics across views. Multi-view training is crucial for decision-making tasks with occlusions, such as multi-object manipulation (Haramati et al., 2024; Qi et al., 2025), where an object

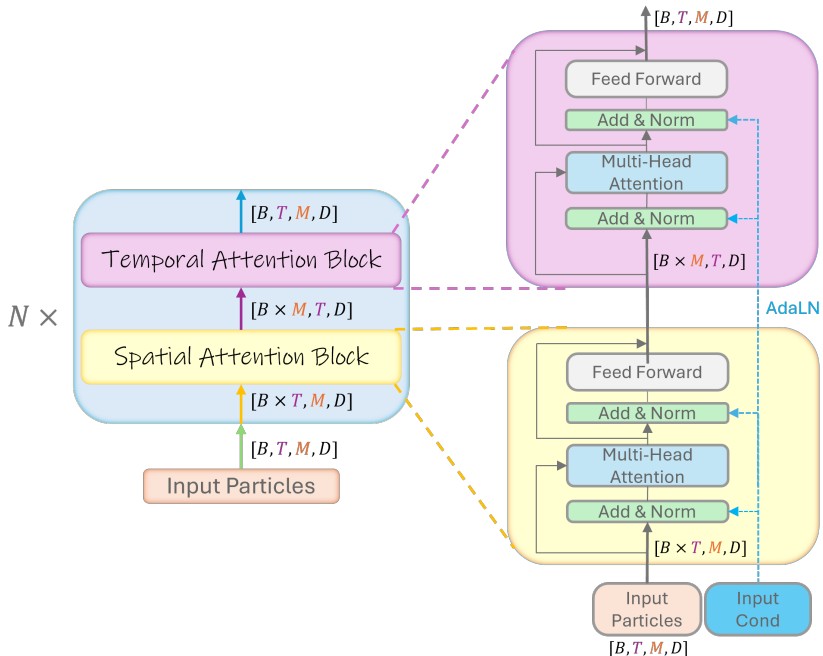

Figure 14: Spatio-temporal transformer block. It consists of (1) a spatial block that independently processes all $M$ particles at each timestep, with shape $[B \times T, M, D]$, and (2) a temporal block that captures each particle's temporal dependencies across the full horizon, with shape $[B \times M, T, D]$. Here, $B$ is batch size, $T$ is temporal horizon, $M$ is number of particles, and $D$ is the embedding dimension.

hidden in one view may be visible in another, enabling agents to form a more complete representation of the scene. To achieve this, images from $V$ views are each encoded into $M$ particles, which are then concatenated into a single set of $V \cdot M$ particles. Each particle is augmented with a view embedding to indicate its origin. The latent action and dynamics modules process the entire multi-view particle set jointly, allowing particles from different views to attend to each other and integrate information across viewpoints.

### A.4.6 Loss $\mathcal{L}$

Latent Particle World Models are trained as a variational autoencoder (VAE), with the objective of maximizing the temporal evidence lower bound (ELBO), as in prior VAE-based video prediction models (Denton & Fergus, 2018; Lin et al., 2020a; Daniel & Tamar, 2024). The ELBO decomposes into a reconstruction term and a KL-divergence term between the posterior particle distributions and the prior predicted by the dynamics model. We further distinguish between two regimes: a *static* ELBO, where the KL-divergence is computed against fixed priors for the first timestep (equivalent to the single-image DLP objective), and a *dynamic* ELBO, where the prior is given by the autoregressive dynamics predictions. Formally,

$$\mathcal{L}_{\text{LPWM}} = -\sum_{t=0}^{T-1} ELBO(x_t = I_t) = \mathcal{L}_{\text{static}} + \mathcal{L}_{\text{dynamic}}. \quad (2)$$

We next detail the formulation of the static and dynamic components.

**Static ELBO:** Building on DLPv2 (Daniel & Tamar, 2024), given an input image $x \in \mathbb{R}^{C \times H \times W}$, the loss in DLPv3 is defined as:

$$\mathcal{L}_{\text{static}} = \mathcal{L}_{\text{rec}}(x, \hat{x}) + \beta_{\text{KL}} \mathcal{L}_{\text{KL}}(z) + \beta_{\text{reg}} \mathcal{L}_{\text{reg}}(z_t), \quad (3)$$

where $\hat{x}$ is the reconstructed image, $\mathcal{L}_{\text{rec}}(x, \hat{x})$ is the reconstruction loss, $z = \left[ \{z_{\text{fg}}^m\}_{m=0}^{M-1}, z_{\text{bg}} \right]$ are the posterior particle distribution parameters, $\mathcal{L}_{\text{KL}}(z)$ is the KL-divergence between the poste-

rior and fixed priors, $z_t$ is the *transparency* attribute of each particle, $\mathcal{L}_{\mathrm{reg}}(z_t)$ is a regularization loss applied over the transparency values, and $\beta_{\mathrm{KL}}, \beta_{\mathrm{reg}}$ are scalar hyperparameters balancing the losses (Higgins et al., 2017). For the single-image setting (i.e., no temporal dynamics), $\mathcal{L}_{\mathrm{static}}$ is the only objective. Below we detail the KL-divergence and regularization terms.

**KL-divergence loss $\mathcal{L}_{\mathrm{KL}}$:** For all $M$ particles, we compute the KL-divergence of each attribute distribution with respect to its fixed prior, except for the background particle which only has visual features. *In DLPv3, we adopt the* masked *KL-divergence (Lin et al., 2020a), where the mask is defined by the transparency attribute $z_t$ (e.g., particles with $z_t = 0$ do not contribute to the KL term):*

$$\mathcal{L}_{\mathrm{KL}}(z) = \sum_{m=0}^{M-1} \left( \sum_{\mathrm{att} \in \{o,s,d\}} \mathrm{KL}(q_\phi(z_{\mathrm{att}}^m \mid x, z_a^m) \,\|\, p_{\mathrm{att}}(z)) \odot z_t^m \right.$$
$$\left. + \mathrm{KL}(q_\phi(z_t^m \mid x, z_a^m)) + \beta_f \, \mathrm{KL}\big(q_\phi(z_f^m \mid x, z_p)\big) \odot z_t^m \right)$$
$$+ \beta_f \, \mathrm{KL}(q_\phi(z_{\mathrm{bg}} \mid x, z_p) \,\|\, p_{\mathrm{bg}}(z)), \tag{4}$$

where $o, s, d$ denote the offset ($z_o$), scale ($z_s$), and depth ($z_d$) attributes, $z_a$ is the keypoint proposal, $z_t$ is the transparency, $z_p = z_a + z_o$ is the final particle position, $z_f$ is the visual features attribute, and $\beta_f$ is a fixed hyperparameter balancing explicit and visual attributes ($\beta_f = 0.01$ in all experiments). Note that the KL for the transparency attribute is *not* masked.

**Transparency regularization $\mathcal{L}_{\mathrm{reg}}$:** *In DLPv3, to prevent the trivial solution where all particles remain active ($z_t = 1$) and sit at patch centers (i.e., a patch-based decomposition), we apply an $L_2$ penalty on transparency:*

$$\mathcal{L}_{\mathrm{reg}} = \sum_{m=0}^{M-1} (z_t^m)^2. \tag{5}$$

This penalty encourages *sparsity* in transparency so that only a subset of particles is active ($z_t > 0$). Since inactive particles do not contribute to reconstruction, the remaining active particles must cover more of the scene and are thereby incentivized to move off patch centers and lock onto salient objects. This yields a more object-centric decomposition and reduces per-particle appearance variance in decoding (e.g., a moving ball is better captured by a single active particle than by several fixed patches). In practice, we set $\beta_{\mathrm{reg}} = \beta_{\mathrm{KL}}$.

**Dynamic ELBO.** The dynamic component of the ELBO consists of three terms: the frame reconstruction loss, the particle dynamics KL, and the context KL:

$$\mathcal{L}_{\mathrm{dynamic}} = \sum_{t=1}^{T-1} \left( \mathcal{L}_{\mathrm{rec}}(x_t, \hat{x}_t) \right.$$
$$+ \beta_{\mathrm{dyn}} \, \mathrm{KL}\big[q_\phi(z^t \mid x_t) \,\|\, p_\xi(z^t \mid z^{0:t-1}, z_c^{0:t-1})\big]$$
$$\left. + \beta_{\mathrm{ctx}} \, \mathrm{KL}\Big[p_\psi^{\mathrm{inv}}(z_c^t \mid z^{0:t+1}) \,\|\, p_\psi^{\mathrm{policy}}(z_c^t \mid z^{0:t})\Big] \right). \tag{6}$$

where $\mathcal{L}_{\mathrm{rec}}$ is the reconstruction error as defined in the static ELBO. Here, $z$ denotes all particles and their attributes (including the background particle), while $z_c$ denotes the latent actions. The coefficients $\beta_{\mathrm{dyn}}$ and $\beta_{\mathrm{ctx}}$ weight the two KL terms (in practice we set $\beta_{\mathrm{ctx}} = \beta_{\mathrm{dyn}}$). Note that for brevity we omitted the particle index $m$; in practice, the summation is carried out over all $M$ particles for both the dynamics and context losses.

For the particle dynamics KL, we adopt the same ***masked*** formulation as in the static ELBO, without distinguishing between explicit attributes and visual features. For the context KL, however, we do not apply masking: latent actions must also capture discrete events where particles switch between inactive ($z_t = 0$) and active ($z_t = 1$). Optimizing this loss end-to-end regularizes the posterior particle distributions to remain predictable under the learned particle dynamics and latent action models. Intuitively, the context KL enforces agreement between the inverse model of latent actions

(which infers actions from observed transitions) and the policy prior (which proposes actions given the current state), ensuring coherent action-conditioned dynamics.

Finally, we specify the choice of priors and reconstruction losses used in practice.

**Priors:** For the fixed static prior distribution parameters, we define means and covariances for Gaussian distributions over all attributes and visual features, and $(a, b)$ in the Beta distribution for the transparency attribute. These are treated as hyperparameters and detailed in Section A.9.

**Reconstruction objective:** We use either the standard pixel-wise mean squared error (MSE) for simulated environments, or an LPIPS-based $L_2$ perceptual loss (Hoshen et al., 2019) for real-world datasets. When using LPIPS, the total reconstruction loss is the sum of the pixel-wise MSE and a VGG-feature-wise MSE, with the LPIPS feature loss weighted by $\gamma = 0.1$. Formally,

$$\mathcal{L}_{\text{rec}} = \begin{cases} \|x - \hat{x}\|_2^2, & \text{for simulated environments,} \\ \|x - \hat{x}\|_2^2 + \gamma \|\phi(x) - \phi(\hat{x})\|_2^2, & \text{for real-world datasets,} \end{cases}$$

where $\phi(\cdot)$ denotes VGG features.

## A.5 POLICY LEARNING WITH LATENT PARTICLE WORLD MODELS

Pre-training a Latent Particle World Model (LPWM) enables the extraction of rich latent dynamics from large-scale, actionless video datasets. Once paired video-action data becomes available, such pre-trained LPWMs can be leveraged for downstream policy learning. In this work, we demonstrate this ability in goal-conditioned settings, where the goal can be specified with an image or a language instruction.

The key idea is to use the CONTEXT module ($\mathcal{K}_\psi$) to learn a mapping from per-particle latent actions $\{z_c^{m,t}\}_{m=0}^M$ to the ground-truth (GT) environment actions $a_t$. While the latent actions encode the transition from latent state $z^t$ to $z^{t+1}$, LPWM produces a latent action per particle; thus, the mapping network must first aggregate information across particles to predict a single global action.

To address this, we design the mapping network $m_\omega$ as an *attention pooling* network (Dosovitskiy et al., 2020; Haramati et al., 2024), implemented as a compact two-layer transformer. This enables the model to adaptively pool the per-particle latent actions before regressing the global action.

**Training procedure:** Given a dataset of paired trajectories $(I_{0:T}, a_{0:T-1})$, we encode the image sequence with a frozen, pre-trained unconditional LPWM to obtain latent actions $\{z_c^{m,t}\}_{m=0}^M$ using the latent inverse dynamics head $p_\psi^{\text{inv}}(z_c^t \mid z^{\leq T})$. These are projected to the transformer's inner dimension $D$. At each timestep, we concatenate a learned action token [ACT] to the particle dimension, forming an input of shape $[B, T, M + 2, D]$, where $B$ is the batch size. Here, the $M$ particles correspond to the $M$ foreground particles, the additional one represents the background particle, and the extra token is the learned action token.

The mapping network regresses the global action from the output corresponding to the action token:

$$\hat{a}_t = m_\omega\big([\{z_c^{m,t}\}_{m=0}^M, [\text{ACT}]_t]\big)_{M+1, t}$$

using the $L_1$ loss:

$$l = \|a_t - \hat{a}_t\|_1.$$

**Inference and planning:** For deployment, given an execution horizon of $k$ actions and a goal $g$, we unroll LPWM $k + 1$ steps autoregressively, generating a trajectory of particles and their corresponding $k$ latent actions. These latent actions are mapped to global actions using the trained mapping network, which are then executed sequentially in the target environment.

An important detail is that for each step, the next-state particles are generated by first sampling a latent action from the CONTEXT module's latent policy head $p_\psi^{\text{policy}}(z_c^t \mid z^{\leq t}, c_t = z_g)$, and then applying the DYNAMICS module. Notably, we empirically found that directly using the latent policy outputs for mapping degrades downstream performance; the mapping network performs best when evaluated on the outputs of the latent inverse module, as this matches the distribution seen during training. The difference may be due to distribution mismatch or higher noise from the latent policy predictor—a question we leave for future investigation.

A high-level PyTorch-style code is provided in Figure 15.

```python
1   # Training loop
2   for (I_seq, a_seq) in dataset:
3       with torch.no_grad():
4           # Encode sequence to latent actions with frozen LPWM inverse dynamics
5           z_c_seq = LPWM(I_seq)  # shape: [B, T, M+1, D_latent]
6
7       # Project latent actions and concatenate learned [ACT] token
8       inputs = concat(z_c_seq, repeat_learned_act_token(T))  # shape: [B, T, M+2, D]
9
10      # Predict global action from [ACT] token output
11      a_pred = mapping_network(inputs)[:, :, -1, :]
12
13      # Compute L1 loss against ground truth actions
14      loss = L1(a_pred, a_seq)
15      loss.backward()
16      optimizer.step()
17  # ----------------- #
18
19  # Inference Loop
20  obs = env.reset()
21  goal = get_goal() # image or language
22  z_particles = encoder(obs)   # Initial particle states
23  z_goal = encode_goal(goal)
24
25  # Storage for predicted particle states during unroll
26  particle_trajectory = [z_particles]
27
28  # Unroll LPWM for (plan_horizon + 1) steps
29  for _ in range(plan_horizon):
30      # Sample latent action from latent policy prior given current particles
31      z_c = ctx.latent_policy(z_particles, z_goal)
32
33      # Predict next particle state given current state and latent action
34      z_particles = dyn(z_particles, z_c)
35
36      # Append predicted particles to trajectory
37      particle_trajectory.append(z_particles)
38
39  # Convert full particle sequence (batch) to latent actions in one call
40  latent_actions = ctx.inverse_dynamics_batch(particle_trajectory)  # outputs plan_horizon
        ↪ latent actions
41
42  # Concatenate learned [ACT] token for mapping network input
43  mapping_inputs = concat(latent_actions, repeat_learned_act_token(plan_horizon))
44
45  # Predict global action sequence from mapping network
46  action_sequence = mapping_network(mapping_inputs)
47
48  # Execute the full predicted action sequence in environment
49  env.step(action_sequence)
```

Figure 15: PyTorch-style code for policy learning with Latent Particle World Models.

## A.6 EXTENDED RELATED WORK

In this section, we provide a broad overview of related literature to situate Latent Particle World Models (LPWM) within the landscape of object-centric representation learning. LPWM is, to the best of our knowledge, the first self-supervised object-centric model capable of being trained only from videos, optionally supporting multi-view training, and enabling various forms of conditioning-including action, language, and goal-image inputs. Since there are currently no other models with the same combination of capabilities, we review several adjacent and complementary lines of work to establish our contributions.

**General video prediction and latent world models:** Traditional video prediction models operate by encoding visual observations into a compact latent space, predicting future latents with a recurrent (or, more recently, autoregressive) dynamics module, and decoding these latents back into pixel space. The notion of a "world model" commonly refers to extensions of this pipeline that support action conditioning and, in some cases, reward modeling. Early methods (Finn et al., 2016a; Ebert et al., 2017b; Villegas et al., 2017; Lee et al., 2018; Denton & Fergus, 2018; Ha & Schmidhuber, 2018; Hafner et al., 2020a) learned such representations using convolutional encoders and RNN-based dynamics to capture scene dynamics in the latent space. More recent work improves

long-horizon prediction robustness using discrete latent variables (Walker et al., 2021; Hafner et al., 2020b; 2023) or focuses on scaling up through hierarchical modeling and larger architectures (Villegas et al., 2019; Wu et al., 2021a; Wang et al., 2022). A core limitation of all these approaches is that they model scene-level dynamics holistically, extracting representations that describe the entire frame at once without explicitly decomposing the scene into objects. This often results in blurry predictions or object disappearance during longer rollouts (Wu et al., 2022b; Daniel & Tamar, 2024). To address these issues and improve sample quality, a number of recent works have incorporated self-attention mechanisms for video dynamics (Nash et al., 2022; Yu et al., 2022; Yan et al., 2021; Zhang et al., 2023; Micheli et al., 2023; 2024; Dedieu et al., 2025), sometimes extending to richer forms of conditioning such as language instructions (Cen et al., 2025; Nematollahi et al., 2025). Despite their advances in visual fidelity, these methods still lack explicit object-centric modeling, and are consistently outperformed by models with object-level inductive biases on tasks involving complex interactions (Wu et al., 2022b; Zhang et al., 2025; Qi et al., 2025). A further trend involves world models based on video diffusion (Alonso et al., 2024; Yang et al., 2023; 2024a; Zhu et al., 2024; Yu et al., 2025), which achieve state-of-the-art generative fidelity but remain computationally intensive and, in their current form, forgo object-centric structure entirely in favor of scaling with model and dataset size.

**Keypoint-based unsupervised video prediction:** A distinct line of research aims to represent video dynamics through keypoint-based latent representations. Early works, such as Kim et al. (2019), combine unsupervised KeyNet (Jakab et al., 2018) keypoint detection with class-guided video prediction using a recurrent adversarial conditional VAE. Similarly, Minderer et al. (2019) and Gao et al. (2021) employ KeyNet for learning keypoints and use a variational RNN prior to model stochastic dynamics, with the latter mapping predicted keypoints onto a discrete grid for more robust long-term prediction. Although these methods successfully leverage keypoints for video structure, they do not explicitly capture object properties or interactions; visual features are extracted directly from feature maps rather than being represented as random latent variables. As a result, prediction quality often suffers from blurriness and object disappearance in long rollouts (Daniel & Tamar, 2022a; 2024). More recent approaches take steps toward modeling interactions: V-CDN (Li et al., 2020) detects unsupervised keypoints using Transporter (Kulkarni et al., 2019), constructs a causal interaction graph, and predicts video outcomes using an Interaction Network (Battaglia et al., 2016). The original Deep Latent Particles (DLP) framework (Daniel & Tamar, 2022a) introduced particle-based video prediction on real data but was limited by its simple graph neural network, which struggled with complex interactions. The subsequent DDLP (Daniel & Tamar, 2024) advanced the framework to object-centric latent particles with richer attributes, enabling more nuanced video predictions. Compared to earlier keypoint-based approaches, DDLP (Daniel & Tamar, 2024) more closely resembles our model, as it captures both rich object attributes and intricate object interactions in video prediction. However, DDLP's reliance on particle tracking and its limitations in handling stochasticity motivate our advances; the present work addresses these challenges while extending the framework toward fully self-supervised world modeling and broader conditioning capabilities, as discussed next.

**Unsupervised object-centric latent video prediction and world models:** Unsupervised methods for object-centric video prediction build latent dynamics over decomposed scene elements, using one of three main paradigms: patch-based, slot-based, or particle-based representations.

Patch-based approaches (e.g., RSQAIR (Stanić & Schmidhuber, 2019), SPAIR (Crawford & Pineau, 2019), SPACE (Lin et al., 2020b), SCALOR (Jiang et al., 2019), G-SWM (Lin et al., 2020a), STOVE (Kossen et al., 2019)) represent objects with local "what", "where", "depth," and "presence" latent attributes, and typically model the joint latent dynamics by RNN-based modules. Later works such as SCALOR and G-SWM incorporated explicit interaction modules to capture object-object physics in prediction. Importantly, GATSBI (Min et al., 2021), an extension of the above with a separate keypoints module, stands out as a patch-based model that can be considered a rudimentary action-conditioned world model, since it predicts scene evolution in response to agent actions. However, unlike particle-based models, keypoints in GATSBI serve only to localize the "agent" in the scene and are not directly part of the object latent representation—most objects and background are discovered through separate modules, and the keypoint module merely distinguishes agent from non-agent entities. As a result, the full object representation remains patch-based rather than explicit keypoint- or particle-based. The patch-based typically require post-hoc or rule-based matching of object proposals across frames for temporal consistency. This reliance on frame-to-frame matching

and the unordered nature of their object pose a scalability challenge to complex or real-world video datasets.

Slot-based approaches (Burgess et al., 2019; Locatello et al., 2020; Greff et al., 2019; Engelcke et al., 2019; 2021; Kipf et al., 2021; Singh et al., 2022b; Kabra et al., 2021; Singh et al., 2021; 2022a; 2023; Sajjadi et al., 2022; Weis et al., 2021; Veerapaneni et al., 2020) typically represent scenes as a set of slots: permutation-invariant latent vectors encoding spatial and appearance information for objects. These approaches generally adopt a two-stage training strategy: a slot decomposition is first learned independently in a self-supervised manner, followed by a separate dynamics model trained on the inferred slots using recurrent models (Zoran et al., 2021; Nakano et al., 2023) or Transformers (Wu et al., 2022b; Villar-Corrales et al., 2023; Song et al., 2024). While recent extensions incorporate conditioning on language (Villar-Corrales et al., 2025; Wang et al., 2025a; Jeong et al., 2025) or latent actions (Villar-Corrales & Behnke, 2025), these models remain fundamentally limited by the quality and stability of the underlying slot decomposition. In practice, slot-based methods suffer from inconsistent decompositions, blurry predictions, and convergence issues, and recent research (Seitzer et al., 2023; Didolkar et al., 2024; Gong et al., 2025; Kakogeorgiou et al., 2024; Jukić et al., 2025) focuses on stabilizing and scaling them, leaving open questions for robust long-term dynamics and world modeling.

Particle-based approaches, initiated by DLP (Daniel & Tamar, 2022a) and advanced by DDLP (Daniel & Tamar, 2024), provide compact, interpretable object representations using keypoint-based latent particles with extended attributes. DDLP jointly trains a Transformer dynamics model and the particle representation, allowing stable object-centric decomposition and improved modeling of complex scenes. However, DDLP relies on particle tracking and sequential encoding, which restricts parallelization and stochasticity. Our proposed LPWM model is a direct extension to this lineage. LPWM eliminates the need for explicit tracking, enabling parallel encoding of all frames, trains end-to-end, and integrates a latent action distribution for stochastic world modeling. This allows the model to capture transitions such as object occlusion, appearance, or random movements (e.g., agents or grippers), and supports comprehensive conditioning via actions, language, or goal images–advancing particle-based modeling to the world model regime and addressing unsolved limitations of previous work.

**Video prediction and world models with latent actions:** To enable learning controllable or playable environments purely from videos, several works propose the use of *latent actions*-global latent variables that model the dynamic transition between consecutive frames. Models such as CADDY (Menapace et al., 2021; 2022) and Genie (Bruce et al., 2024; Savov et al., 2025) learn *discrete* latent actions by quantizing the output of an inverse dynamics module. These latent actions condition a dynamics model to generate subsequent frames. Crucially, these approaches use a two-stage training process: first, the latent action module is trained, then the conditioned dynamics module. During inference, users select latent actions from a learned codebook to generate video sequences. AdaWorld (Gao et al., 2025) proposes a continuous analog without quantization, substituting quantization with strong KL regularization on the latent action distribution. This enables more flexible and smooth latent action representations. PlaySlot (Villar-Corrales & Behnke, 2025), the method most similar to ours in this category, augments slot-based object-centric prediction (OCVP (Villar-Corrales et al., 2023)) with a discrete global latent action module akin to CADDY, showcasing the benefits of object-centric decomposition for controllable video modeling. In contrast, our particle-based LPWM learns *continuous*, *per-particle* latent actions trained end-to-end jointly with the dynamics module. This design captures stochastic dynamics across multiple entities simultaneously. Furthermore, LPWM regularizes latent actions using a learned latent policy, enabling stochastic sampling of latent actions at inference without external intervention, thereby supporting stochastic video generation. Additionally, unlike PlaySlot, LPWM's latent action module supports multiple conditioning modalities—including goal-conditioning—making it readily applicable for post-hoc policy learning and control, as demonstrated in our experiments.

**Decision-making with video inverse dynamics and latent actions:** Recent works have increasingly focused on learning policies from videos by leveraging inverse dynamics modeling (IDM) or latent action representations. ILPO (Edwards et al., 2019) learns discrete latent actions via forward dynamics under the assumption of a known action set, then maps these latent actions to ground-truth (GT) actions for behavioral cloning (BC). Seer (Tian et al., 2024) jointly trains image prediction and GT action prediction via inverse modeling, without a latent action bottleneck, and effectively supports language-conditioned BC. LAPO (Schmidt & Jiang, 2023) combines dis-

crete latent actions learned via vector quantization (VQ) with policy learning; an action decoder is jointly trained alongside an online reinforcement learning agent to map latent actions to GT actions. LAPA (Ye et al., 2025) performs large-scale VQ-based latent action pretraining, which serves as the objective for vision-language model policy training, and fine-tunes for GT action mapping, with AMPLIFY (Collins et al., 2025) extending this by replacing latent actions with quantized keypoint tracks. DreamGen (Jang et al., 2025) further extends LAPA with diffusion-based objectives. AdaWorld (Gao et al., 2025) pre-trains large-scale autoregressive world models with continuous latent actions for downstream planning, while VideoWorld (Ren et al., 2025) trains an autoregressive discrete latent action model producing latent plans and frame-level IDM over decoded plans. Latent Diffusion Planning (Xie et al., 2025) and VILP (Xu et al., 2025) learn diffusion-based planners coupled with inverse dynamics modules. Similarly, Video Prediction Policy (Hu et al., 2025) fine-tunes a text-based large video diffusion model for plan generation, then learns a diffusion-based policy via inverse dynamics. DynaMo (Cui et al., 2024) pre-trains image representations with paired inverse and forward dynamics self-supervised objectives for policy learning over these representations. In contrast to these approaches, LPWM is an object-centric world model that integrates latent action learning directly with dynamics, producing per-entity latent actions that naturally accommodate multiple interacting objects. Its latent policy further enables effective post-hoc mapping to GT actions and direct application to behavioral cloning and control tasks, distinguishing it from predominantly global or multi-stage latent action frameworks.

**Decision-making with object-centric representations:** As object-centric representations have matured, they have been increasingly incorporated into decision-making pipelines, demonstrating strong performance on multi-object tasks that require complex reasoning and interaction. SMORL (Zadaianchuk et al., 2022) leverages patch-based object-centric representations in online RL, showing that structured perception improves sample efficiency and enables control in environments with multiple entities. ECRL (Haramati et al., 2024) and EC-Diffuser (Qi et al., 2025) employ DLP-based (particle-centric) representations, integrating them into online RL (ECRL) or imitation learning with diffuser-based policies (EC-Diffuser). These results provide clear evidence that object-centric models facilitate efficient policy learning and handle multi-object interaction challenges. Complementary lines of work adapt slot-based world models for decision-making. FOCUS (Ferraro et al., 2025; 2024), SOLD (Mosbach et al., 2024), and Dyn-O (Wang et al., 2025b) augment online RL frameworks like Dreamer (Hafner et al., 2020b) with slot-based object decomposition, yielding improvements in simulated environments featuring a limited number of objects. OC-STORM (Zhang et al., 2025) extends STORM (Zhang et al., 2023) by combining a transformer-based dynamics module with object masks derived from supervised segmentation, thus relying on labeled inputs for decomposition. SegDAC (Brown & Berseth, 2025) and OCAAM (Rubinstein et al., 2025) similarly use deep RL over masked inputs, leveraging externally supervised segmentation models. Some recent efforts bridge object-centric decomposition with latent action learning: Klepach et al. (2025) trains latent actions on top of pre-trained slot-based representations, mapping these actions to ground-truth policies via imitation learning. In contrast, LPWM is a fully self-supervised, object-centric world model: it builds directly on DLP, is trained end-to-end from pixels, and learns per-object latent actions as part of its joint dynamics training—without requiring supervised segmentation or decoupled vision/policy phases, with object masks emerging as a natural result of its reconstruction objective rather than any external supervision. Finally, LPWM supports post-hoc multi-object imitation learning and behavioral cloning in complex scenes.

Table 4 summarizes the various video prediction world modeling approaches across key dimensions.

## A.7    Datasets and Environments Details

We provide detailed descriptions of all datasets used in this paper. Datasets are characterized by their properties: real-world or simulated origin, nature of dynamics—deterministic (dominated by physics, no external actions) or stochastic (external signals such as agent actions or camera motion)—and interaction density. Some datasets feature dense interactions, where object interactions are frequent and most sequences include them, while others are sparse, with less frequent or delayed interactions, and some sequences may contain no interactions (Daniel & Tamar, 2024).

`OBJ3D`: A simulated 3D dataset featuring dense interactions and deterministic dynamics, introduced by Lin et al. (2020a). It consists of CLEVR-like objects (Johnson et al., 2017) in 100-frame videos of $128 \times 128$ resolution, where a randomly colored ball rolls towards multiple objects in the scene

| Model | Obj.-Centric | Latent Actions | Action Cond. | Text Cond. | Image-Goal Cond. | End-to-End | Multi-View | Stochastic | Dyn. Module |
|---|---|---|---|---|---|---|---|---|---|
| CDNA/SNA/SVG (Finn et al., 2016a) | – | – | ✓ | – | – | ✓ | – | – | RNN |
| Dreamer/D2/D3 (Hafner et al., 2020a) | – | Discrete | ✓ | – | – | ✓ | – | ✓ | RNN |
| CADDY (Menapace et al., 2021) | – | Discrete | ✓ | – | – | – | – | ✓ | RNN |
| Genie (Bruce et al., 2024) | – | Discrete | ✓ | – | – | – | – | ✓ | Transformer |
| UniSim (Yang et al., 2023) | – | – | ✓ | ✓ | Varies | ✓ | varies | ✓ | Diffusion |
| Diamond/GameFactory (Alonso et al., 2024) | – | – | ✓ | – | – | ✓ | varies | ✓ | Diffusion |
| VideoGPT (Yan et al., 2021) | – | – | – | – | – | ✓ | – | ✓ | Transformer |
| SCALOR (Jiang et al., 2019) | Patch | – | – | – | – | ✓ | – | ✓ | RNN |
| G-SWM (Lin et al., 2020a) | Patch | – | – | – | – | ✓ | – | ✓ | RNN |
| STOVE (Kossen et al., 2019) | Patch | – | – | – | – | ✓ | – | ✓ | RNN |
| OCVT (Wu et al., 2021b) | Patch | – | ✓ | – | – | – | – | – | Transformer |
| GATSBI (Min et al., 2021) | Patch+Keypt | – | – | – | – | ✓ | – | ✓ | RNN |
| V-CDN (Li et al., 2020) | Keypt+Graph | – | – | – | – | – | – | ✓ | GNN |
| PARTS (Zoran et al., 2021) | Slots | – | – | – | – | ✓ | – | – | RNN |
| STEDIE (Nakano et al., 2023) | Slots | – | – | – | – | ✓ | – | – | RNN |
| SlotFormer (Wu et al., 2022b) | Slots | – | – | – | – | – | – | – | Transformer |
| OCVP (Villar-Corrales et al., 2023) | Slots | – | – | – | – | – | – | – | Transformer |
| TextOCVP (Villar-Corrales et al., 2025) | Slots | – | – | ✓ | – | – | – | – | Transformer |
| SOLD (Mosbach et al., 2024) | Slots | – | ✓ | – | – | – | – | – | Transformer |
| PlaySlot (Villar-Corrales & Behnke, 2025) | Slots | Discrete | – | – | – | – | – | – | Transformer |
| DLP (Daniel & Tamar, 2022a) | Particles | – | – | – | – | – | – | – | GNN |
| DDLP (Daniel & Tamar, 2024) | Particles | – | – | – | – | ✓ | – | – | Transformer |
| LPWM (Ours) | Particles | Cont. (per) | ✓ | ✓ | ✓ | ✓ | ✓ | ✓ | Transformer |

Table 4: Comparison of video prediction and world modeling approaches across key dimensions. Models are grouped by representation category: holistic, patch/object-centric, slot/object-centric, and particle/object-centric. AR: autoregressive; GNN: graph neural network. "Image-Goal Cond." is image-goal conditioning support.

center, causing collisions. The dataset includes 2,920 training episodes, 200 validation, and 200 test episodes.

`PHYRE`: A simulated 2D dataset featuring sparse interactions and deterministic dynamics, designed for physical reasoning (Bakhtin et al., 2019). We use the BALL-tier tasks in the ball-within-template setting, where tasks are solved if a user-placed ball satisfies specific conditions (e.g., touching a wall, floor, or object). Data consists of $128 \times 128$ frames generated from rollouts of all tasks except for tasks $[12, 13, 16, 20, 21]$, which contain substantial distractions. The dataset contains 2,574 training episodes, 312 validation, and 400 test episodes.

`Mario`: A simulated 2D dataset with stochastic dynamics and dense interactions, introduced by Smirnov et al. (2021). It consists of expert gameplay videos of Super Mario Bros downloaded from YouTube, featuring Mario traversing multiple levels. The videos include moving camera views with new objects and enemies appearing dynamically. The dataset contains 217 training trajectories and 25 test trajectories, each consisting of 100 frames with resolution $128 \times 128$. For FVD evaluation, we sample 100 trajectories for each video in the test set.

`Sketchy`: A real-world robotic dataset introduced by Cabi et al. (2019), featuring a robotic gripper interacting with diverse objects. It has stochastic dynamics with sparse interactions. We focus on the `stack_green_on_red` task, which includes 198 expert and 3,241 rollout trajectories. The dataset is split into 80% training, 10% validation, and 10% test. Each trajectory is truncated to the first 70 frames, resized to $128 \times 128$, and contains labeled actions enabling action-conditioned training.

`BAIR`: A real-world robotic dataset introduced by Ebert et al. (2017a), featuring a robotic gripper manipulating diverse objects under a random play policy. The dataset exhibits stochastic dynamics and dense interactions, containing 43,264 training and 256 test trajectories at $64 \times 64$ resolution. For evaluation of FVD we follow the standard procedure of sampling 100 trajectories for each video in the test set (a total of 25,600 of generated videos). For $128 \times 128$ resolution training we use the high-resolution version of the dataset introduced in Menapace et al. (2021), which contains 42,880 train trajectories, 1,152 for validation and 128 for test.

`Bridge`: A real-world robotic dataset introduced by Walke et al. (2023), featuring expert demonstrations of a WidowX robotic arm performing tasks guided by natural language instructions. It exhibits stochastic dynamics and dense interactions. The dataset contains 25,460 training and 3,475 test trajectories, with episodes of varied lengths, all resized to $128 \times 128$ resolution.

`LanguageTable`: A real-world tabletop robotic dataset introduced by Lynch et al. (2023), featuring language-guided, action-annotated demonstrations of complex relational object arrangements based on shape, color, and relative position. The dataset exhibits stochastic dynamics with dense interactions and contains 179,976 episodes of variable length, resized to $128 \times 128$ resolution. We use an 80% training, 10% validation, and 10% test split.

`PandaPush`: A simulated 3D robotic environment introduced in ECRL (Haramati et al., 2024) and used by EC-Diffuser (Qi et al., 2025) for goal-conditioned imitation learning. The task involves Isaac Gym-based (Makoviychuk et al., 2021) tabletop manipulation using a Franka Panda arm to push colored cubes to a goal configuration specified by an image. We utilize the same offline two-view image dataset collected in EC-Diffuser, comprising approximately 9,000 episodes with 1–3 cubes each, and around 30–40 $128 \times 128$ frames per episode from each view.

`OGBench-Scene`: A simulated 3D environment and dataset from the offline goal-conditioned reinforcement learning benchmark OGBench (Park et al., 2025). Specifically, we use the *Visual Scene* environment and the associated "play" dataset, which features non-goal-directed interactions of a 6-DoF UR5e robot arm with various objects in a tabletop setting. The dataset includes 1,000 training and 100 validation trajectories, each containing 1,000 transitions, recorded at a resolution of $64 \times 64$.

### A.8 BASELINE DETAILS

**OCVP and PlaySlot:** We use the official implementations (Villar-Corrales, 2025) of OCVP and PlaySlot and adapt the dynamic modules sizes to match LPWM, alongside modifying the CNN components for compatibility with $128 \times 128$ input resolution. Both models are trained in multiple stages, beginning with slot-based decomposition using SAVi (Elsayed et al., 2022). Downstream

video prediction performance is highly dependent on the quality of this initial decomposition. As noted in previous works (Daniel & Tamar, 2024; Didolkar et al., 2024), SAVi can fail to assign distinct objects to separate slots and may require repeated runs with identical hyperparameters to achieve satisfactory results. While we primarily adhere to recommended hyperparameters, slot assignments are sometimes ambiguous, with multiple objects per slot and occasional blurry reconstructions.

**DVAE:** We implement a non-object-centric, patch-based dynamics VAE (DVAE) world model adapted from LPWM. In DVAE, "particles" correspond to fixed-grid patch embeddings, where the number of patches matches LPWM's particle count, $M$. The baseline architecture preserves the same transformer backbone and parameter budget as LPWM, and supports identical conditioning modes (e.g., actions, language, image goals). However, DVAE does not model explicit object attributes, relying instead on spatially organized patch features. This approach is analogous to patch-based tokenization schemes commonly used in large-scale video generation (Yan et al., 2021; Yang et al., 2024b), but the patch embeddings are learned end-to-end without pretraining or quantization, similar to LPWM's particle learning. To compensate for the lack of object-centric structure, we increase the latent dimension of each patch embedding. Patch extraction follows the standard procedure (Esser et al., 2021), whereby a CNN encoder downsamples input frames by a factor of $f$ until the spatial dimensions are $M = \frac{H}{f} \times \frac{W}{f}$, with $H$ and $W$ being the input height and width. The resulting grid of patch features is used as the input "particle" set for downstream dynamics modeling and video prediction.

### A.9    Hyperparameters and Training Details

**Hyperparameters.** We use the Adam (Kingma & Ba, 2014) optimizer ($\beta_1 = 0.9, \beta_2 = 0.999, \epsilon = 1e-6$) with a constant learning rate of $8e-5$. For all models we use an inner transformer projection dimension of $512$, and the latent actions dimension is set to $d_{ctx} = 7$. The constant prior distribution parameters, reported in Table 6, depend on the patch size used to extract the particles attributes and features. The complete set of the rest of the hyperparameters can be found in Table 5.

**Warmup.** In our training procedure, given a sequence of $T$ frames, we typically apply the static ELBO loss to the first frame and the dynamic ELBO loss to the remaining $T-1$ frames. To facilitate robust learning of the initial particle decomposition, we introduce a warmup stage during the first few iterations, usually corresponding to the first training epoch. In this stage, the static ELBO is applied to the first $T-1$ frames, and only the final frame receives the dynamic ELBO. This warmup provides a strong initialization for the dynamics module and improves overall training stability and downstream performance.

**Burn-in frames.** Previous work (Wu et al., 2022b; Villar-Corrales et al., 2023; Daniel & Tamar, 2024) introduces the concept of *burn-in frames*, where the first $n$ frames in a sequence (typically $4 \leq n \leq 6$) are provided as conditioning inputs to drive dynamics prediction under the assumption of deterministic dynamics. In DDLP, these initial frames are optimized using the static ELBO. In contrast, LPWM does *not* employ burn-in frames, as we assume stochastic dynamics and instead rely on latent actions to drive predictions.

**Stopping criteria.** During training, we track several metrics calculated on the validation sets. Mainly, we save checkpoints for the best validation ELBO value and best validation LPIPS value, where the latent-action-conditioned generated video is compared to the GT video.

**Resources.** All experiments were conducted on various cloud-based computing platforms. For most datasets, training utilized a single NVIDIA A100 or GH200 GPU. For larger-scale datasets such as `LanguageTable`, `Bridge`, and `Panda`, training was performed on 8 GPUs, either A100s or H100s. The training duration varies by dataset size: small to medium datasets typically require a few days to train, while large-scale datasets may take up to two weeks.

### A.10    Additional Experiments and Results

This section presents additional experimental results and further insights complementing the main findings of this paper. In Section 7, we compare our modified DLPv3 model to its predecessors. Section A.10.2 provides further video prediction results, followed by an ablative analysis in Sec-

| Hyperparameter | OBJ3D | PHYRE | Sketchy | Mario | BAIR | Bridge | LangTable | PandaPush | OGBench |
|---|---|---|---|---|---|---|---|---|---|
| Resolution | $128 \times 128$ | $128 \times 128$ | $128 \times 128$ | $128 \times 128$ | $128 \times 128$ | $128 \times 128$ | $128 \times 128$ | $128 \times 128$ (2 Views) | $64 \times 64$ |
| $L$ (# Particles) | 12 | 64 | 30 | 90 | 90 | 50 | 24 | 25 | 24 |
| $M$ (# KP Proposals) | 64 | 256 | 64 | 256 | 256 | 64 | 256 | 64 | 64 |
| $T$ (Training Horizon) | 20 | 15 | 20 | 20 | 16 | 24 | 20 | 10 | 30 |
| Reconstruction Loss | LPIPS | MSE | LPIPS | MSE | LPIPS | LPIPS | LPIPS | MSE | MSE |
| $\beta_{\text{KL}}$ | 0.08 | 0.02 | 0.08 | 0.02 | 0.08 | 0.08 | 0.08 | 0.04 | 0.02 |
| $\beta_{\text{dyn}}$ | 0.2 | 0.05 | 0.2 | 0.05 | 0.2 | 0.2 | 0.2 | 0.1 | 0.05 |
| $\beta_{\text{reg}}$ | 0.08 | 0.02 | 0.08 | 0.02 | 0.08 | 0.08 | 0.0 | 0.04 | 0.02 |
| KP Proposal Patch Size | 16 | 8 | 8 | 16 | 8 | 8 | 8 | 16 | 8 |
| Glimpse Ratio | 0.25 | 0.125 | 0.25 | 0.125 | 0.125 | 0.25 | — | 0.25 | 0.25 |
| $d_{\text{obj}}$ | 4 | 4 | 4 | 5 | 5 | 6 | 5 | 4 | 4 |
| $d_{\text{bg}}$ | 4 | 4 | 4 | 5 | 5 | 6 | 5 | 2 | 2 |
| FG CNN Ch. Mult. | $[1, 4, 8]$ | $[2, 4, 8]$ | $[1, 4, 8]$ | $[1, 4, 8]$ | $[2, 4, 8]$ | $[1, 4, 8]$ | $[2, 4, 8]$ | $[1, 4, 8]$ | $[1, 2, 2]$ |
| BG CNN Ch. Mult. | $[1, 1, 1, 2, 4]$ | $[1, 1, 1, 2, 4]$ | $[1, 1, 1, 2, 4]$ | $[1, 1, 1, 2, 8]$ | $[1, 1, 1, 2, 4]$ | $[1, 1, 1, 2, 4]$ | $[1, 1, 1, 2, 4]$ | $[1, 1, 1, 4, 8]$ | $[1, 1, 2, 2]$ |
| # $\mathcal{K}_\psi$ Layers | 4 | 4 | 4 | 4 | 4 | 6 | 6 | 4 | 4 |
| # $\mathcal{K}_\psi$ Heads | 8 | 8 | 8 | 8 | 8 | 8 | 8 | 8 | 8 |
| # $\mathcal{F}_\xi$ Layers | 6 | 6 | 6 | 6 | 6 | 8 | 8 | 6 | 6 |
| # $\mathcal{F}_\xi$ Heads | 8 | 8 | 8 | 8 | 8 | 8 | 8 | 8 | 8 |
| # Epochs | 16 | 15 | 18 | 200 | 20 | 42 | 50 | 84 | 80 |
| Model Size | 110M | 110M | 110M | 110M | 110M | 147M | 146M | 112M | 103M |
| FLOPs (Inference) | 3153G | 8272G | 3153G | 8272G | 6623G | 4667G | 32715G | 6641G | 4142G |
| FLOPs (Generation) | 16036G | 121642G | 16036G | 121642G | 60731G | 33699G | 185967G | 46660G | 34569G |

Table 5: Hyperparameters across datasets. Base CNN channels count is 32. $\mathcal{K}_\psi$ refers to the Transformer-based DYNAMICS. FLOPs - floating-point operations per second (higher means more computational operations per second). FLOPs (Inference) corresponds to encoding and decoding of 16 frames, and FLOPs (Generation) correspond to one forward rollout of 15 frames conditioned on 1 frame.

| Attribute | Distribution | Parameters (glimpse_ratio = 0.25) | Parameters (glimpse_ratio = 0.125) |
|---|---|---|---|
| Position Offset $z_o$ | Normal, $\mathcal{N}(\mu, \sigma^2)$ | $\mu = 0,\ \sigma = 0.2$ | $\mu = 0,\ \sigma = 0.1$ |
| Scale $z_s$ | Normal, $\mathcal{N}(\mu, \sigma^2)$ | $\mu = \text{Sigmoid}^{-1}(0.25),\ \sigma = 0.3$ | $\mu = \text{Sigmoid}^{-1}(0.125),\ \sigma = 0.15$ |
| Depth $z_d$ | Normal, $\mathcal{N}(\mu, \sigma^2)$ | $\mu = 0,\ \sigma = 1$ | $\mu = 0,\ \sigma = 1$ |
| Transparency $z_t$ | Beta, $\text{Beta}(a, b)$ | $a = 0.01,\ b = 0.01$ | $a = 0.01,\ b = 0.01$ |
| Appearance Features $z_f, z_{\text{bg}}$ | Normal, $\mathcal{N}(\mu, \sigma^2)$ | $\mu = 0,\ \sigma = 1$ | $\mu = 0,\ \sigma = 1$ |

Table 6: Prior distribution parameters for different glimpse (patch) ratios. Glimpses are patches taken around keypoints, where glimpse_ratio $= \frac{\text{glimpse size}}{\text{image size}}$.

tion A.10.3. Finally, Section A.10.4 discusses and analyzes our imitation learning-based decision-making application.

### A.10.1 COMPARISON OF DLPV3, DLPV2, AND DLPV1

We quantitatively evaluate our enhanced DLP variant, DLPv3 (see Section A.4), against the original DLP (Daniel & Tamar, 2022a) and DLPv2 (Daniel & Tamar, 2024) using publicly available implementations (Daniel & Tamar, 2022b; Daniel, 2024). All models are trained in the single-image setting on the `OBJ3D` dataset, with identical particle counts and recommended hyperparameters. Training for each model is terminated when the validation LPIPS score ceases to improve. As presented in Table 7, DLPv3 achieves substantially superior image reconstruction compared to prior versions. Notably, DLPv1 lacks explicit modeling of object attributes and is therefore unable to generate bounding boxes and other attributes that contribute to the performance.

| | OBJ3D | | |
|---|---|---|---|
| | PSNR ↑ | SSIM ↑ | LPIPS ↓ |
| **DLP** | $39.23 \pm 3.33$ | $0.982 \pm 0.009$ | $0.085 \pm 0.018$ |
| **DLPv2** | $41.97 \pm 3.74$ | $0.985 \pm 0.006$ | $0.019 \pm 0.01$ |
| **DLPv3** | $\mathbf{43.87 \pm 4.45}$ | $\mathbf{0.990 \pm 0.006}$ | $\mathbf{0.011 \pm 0.005}$ |

Table 7: DLPv3, DLPv2, and DLP image reconstruction performance comparison in the single-image setting, evaluated on the test set.

### A.10.2 SELF-SUPERVISED OBJECT-CENTRIC VIDEO PREDICTION AND GENERATION

Table 2 demonstrates that LPWM surpasses all baselines on LPIPS and FVD metrics across datasets with stochastic dynamics under all conditioning settings. Notably, Figure 1 illustrates that LPWM effectively preserves *object permanence* over the entire generation horizon, whereas competing methods often suffer from object blurring and deformation. Furthermore, LPWM accurately models complex object interactions which are better aligned with the language instructions, as evidenced by rollouts on various robotic datasets.

We also highlight LPWM's multi-modality sampling capability: by drawing multiple samples from the latent policy starting from the same initial frames and language prompts, LPWM produces diverse and plausible rollouts. Several examples are presented in Figures 16 and in videos available on our project website. Results for datasets with deterministic dynamics are detailed in Table 8.

Regarding our main object-centric slot-based baseline, PlaySlot, objects tend to drift rather than remain static. This likely stems from its use of a global latent action vector that models transitions for all entities collectively, unlike our approach that leverages per-particle latent actions. Slot-based models also suffer from limitations inherent to their slot-decomposition modules, which produce blurry reconstructions and imperfect object decompositions, consistent with prior observations. Moreover, slot models struggle on datasets containing many objects (e.g., `Mario`) due to memory-limited number of slots, whereas LPWM's low-dimensional latent particles effectively scale.

DVAE, our primary non-object-centric baseline, performs competitively on synthetic datasets but falls short on real-world datasets, underscoring the advantages of object-centric representations. In certain cases, DVAE even outperforms PlaySlot, likely because both DVAE and LPWM utilize per-patch latent actions. We note that some datasets involve sparse object interactions, and visual metrics tend to emphasize large entities, which can favor DVAE's performance.

Video rollout examples demonstrating these behaviors are available on our project website: `https://taldatech.github.io/lpwm-web`.

**Representation Inductive Bias versus Model Scale**: To further highlight the advantages of object-centric representations over simply scaling up model size using standard patch-based representations, we train an LPWM model with 100M parameters on the standard video prediction benchmark `BAIR-64` and report its FVD in Table 9. Despite its relatively small size, LPWM achieves performance comparable to many larger video generation models. We attribute this to LPWM's inherent strength in modeling object interactions, which provides a significant advantage over large patch-based models that may generate crisp pixel-level details but struggle with physically plausible interactions (e.g., gripper movements intersecting objects). This demonstrates that the inductive biases encoded through object-centric representations can yield benefits that scale alone cannot easily achieve.

| Dataset | OBJ3D | | | PHYRE | | |
|---|---|---|---|---|---|---|
| Setting | $t:20, c:6, p:44$ | | | $t:15, c:10, p:40$ | | |
| | PSNR↑ | SSIM↑ | LPIPS↓ | PSNR↑ | SSIM↑ | LPIPS↓ |
| **DVAE** | 31.44±5.69 | 0.923±0.05 | 0.085±0.07 | 26.61±6.01 | 0.94±0.04 | **0.047±0.04** |
| **G-SWM** | 31.7±6.2 | 0.924±0.05 | 0.118±0.07 | 24.64±6.25 | 0.93±0.05 | 0.078±0.06 |
| **SlotFormer/OCVP** | 31.2±4.91 | 0.925±0.04 | 0.135±0.05 | 21.26±3.54 | 0.89±0.05 | 0.108±0.05 |
| **DDLP** | 31.29±5.22 | 0.923±0.04 | 0.088±0.06 | 26.98±5.3 | 0.95±0.04 | 0.055±0.04 |
| **LPWM** (Ours) | 31.45±5.47 | 0.926±0.04 | **0.081±0.06** | 26.94±5.88 | 0.95±0.04 | **0.048±0.04** |

Table 8: Quantitative results on video prediction for datasets with deterministic dynamics. $t$ is the training horizon, $c$ is the conditional frames at inference and $p$ is the predicted frames at inference.

| `BAIR-64` ($64 \times 64$) | FVD↓ |
|---|---|
| LVT (Rakhimov et al., 2020) | 125.8 |
| DVD-GAN-FP (Clark et al., 2019) | 109.8 |
| TrIVD-GAN-FP (Luc et al., 2020) | 103.3 |
| VideoGPT (Yan et al., 2021) | 103.3 |
| CCVS (Le Moing et al., 2021) | 99.0 |
| FitVid (Babaeizadeh et al., 2021) | 93.6 |
| MCVD (Voleti et al., 2022) | 89.5 |
| NÜWA (Wu et al., 2022a) | 86.9 |
| RaMViD (Höppe et al., 2022) | 84.2 |
| MAGVIT-B (Yu et al., 2022) | 76 |
| RIVER (Davtyan et al., 2023) | 73.5 |
| CVP (Shrivastava & Shrivastava, 2024) | 70.1 |
| VDM (Ho et al., 2022) | 66.9 |
| MAGVIT-L (Yu et al., 2022) | 62 |
| LPWM (Ours) | 89.4 |

Table 9: Video prediction results on `BAIR-64` ($64 \times 64$) conditioning on one past frame and predicting 15 frames in the future. Table adapted from Shrivastava & Shrivastava (2024).

**Language-conditioned video prediction**: Table 10 provides additional visual metrics for language-conditioned settings. Specifically, we report the PSNR, SSIM and LPIPS when using the language-conditioned posterior latent-actions to reproduce the original trajectory, as opposed to the standard practice of sampling language-conditioned latent-actions from the latent prior where only FVD is applicable.

### A.10.3  ABLATION ANALYSIS

We perform ablation studies to evaluate the impact of key design decisions in LPWM, including latent action type (global vs. per-particle), latent action dimensionality, and positional embedding methods. Using the `Sketchy` dataset and evaluating after 10 training epochs (Table 11), we observe that per-particle latent actions are critical for strong latent-action-conditioned video prediction performance. However, for sampling diversity, global mean-pooling of latent actions yields improved FVD, suggesting benefits from global variables during generation. The model demonstrates

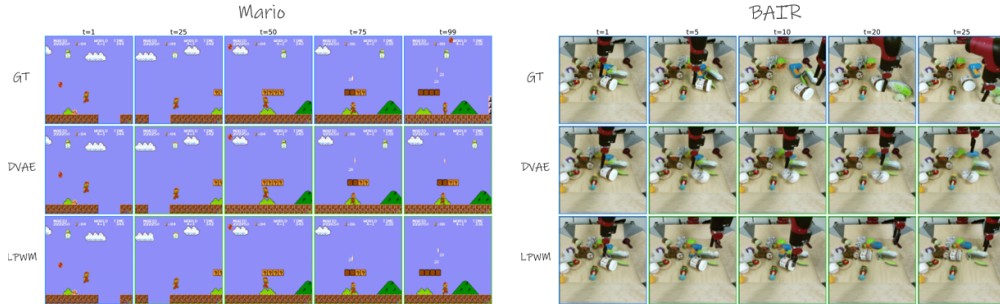

Figure 16: Multi-modal future sampling by LPWM. Starting from the same initial frame, LPWM produces diverse possible future trajectories, illustrated on the `Mario` (left) and `BAIR` (right) datasets.

| Dataset | Bridge-L | | | | LanguageTable-L | | | |
|---|---|---|---|---|---|---|---|---|
| | PSNR↑ | SSIM↑ | LPIPS↓ | FVD↓ | PSNR↑ | SSIM↑ | LPIPS↓ | FVD↓ |
| **DVAE** | 19.37±3.8 | 0.75±0.09 | 0.177±0.078 | 146.85 | 36.0±4.14 | 0.97±0.01 | 0.019±0.01 | 26.78 |
| **LPWM** (Ours) | 26.38±4.1 | 0.87±0.08 | 0.077±0.05 | **47.78** | 36.57±3.01 | 0.97±0.007 | 0.016±0.006 | 15.96 |

Table 10: Quantitative results on language-conditioned (L) video generation. PSNR, SSIM and LPIPS are calculated on latent-action-conditioned video prediction. FVD is reported for stochastic generation by sampling from the latent policy.

robustness to latent action dimension as long as it approximates the effective particle dimension ($< 6 + d_{obj}$, i.e., 10 for `Sketchy`), balancing compression and information retention; our choice of $d_{ctx} = 7$ reflects this trade-off[5]. Finally, adaptive layer normalization (AdaLN) for embedding timestep and particle identity outperforms standard additive positional embeddings, as previously observed (Zhu et al., 2024), albeit with an increased parameter count.

| Ablation Variant | Latent Actions $d_{ctx}$ | Latent Actions Type | Positional Embeddings | PSNR↑ | SSIM↑ | LPIPS↓ | FVD↓ |
|---|---|---|---|---|---|---|---|
| Original | 7 | Per-Particle | Learned AdaLN | 28.55±3.30 | 0.91±0.05 | 0.072±0.03 | 120.32 |
| $d_{ctx} = 1$ | 1 | Per-Particle | Learned AdaLN | 27.71±3.50 | 0.89±0.06 | 0.081±0.03 | 177.64 |
| $d_{ctx} = 3$ | 3 | Per-Particle | Learned AdaLN | 29.08±3.15 | 0.92±0.05 | 0.070±0.03 | 117.46 |
| $d_{ctx} = 10$ | 10 | Per-Particle | Learned AdaLN | 28.97±3.28 | 0.91±0.05 | 0.068±0.03 | 117.54 |
| $d_{ctx} = 14$ | 14 | Per-Particle | Learned AdaLN | 28.81±3.22 | 0.91±0.05 | 0.069±0.03 | 121.02 |
| Global Latent Actions | 7 | Mean Pool | Learned AdaLN | 27.24±3.7 | 0.89±0.07 | 0.087±0.04 | 100.75 |
| Global Latent Actions | 7 | Token Attention Pool | Learned AdaLN | 21.54±4.29 | 0.80±0.11 | 0.176±0.09 | 142.64 |
| Positional Embeddings | 7 | Per-Particle | Learned Additive | 21.54±4.29 | 0.80±0.11 | 0.176±0.09 | 142.64 |

Table 11: Ablation results: impact of latent action dimensions and type, and positional embeddings on LPWM performance. Results are reported on the `Sketchy` dataset after 10 epochs of training. Results do not reflect final performance.

### A.10.4 POLICY LEARNING WITH LATENT PARTICLE WORLD MODELS

This section provides additional details on our decision-making application—learning imitation policies from a pre-trained LPWM, as described in Section 5.2.

`OGBench-Scene`: designed to challenge an agent's long-horizon sequential reasoning through manipulation of diverse objects including cubes, windows, drawers, and buttons. Pressing a button toggles the lock status of associated objects, requiring complex, multi-step planning to arrange objects into target configurations. The baselines are taken directly from the benchmark which includes GCBC (Lynch et al., 2020), GCIQL (a goal-conditioned variant of IQL (Kostrikov et al., 2022)),

---

[5]Results after 10 epochs do not reflect final performance; best performance across datasets achieved with $d_{ctx} = 7$

| Task | VQ-BeT | Diffuser | EIT+BC | EC Diffusion Policy | EC Diffuser | LPWM (Ours) |
|---|---|---|---|---|---|---|
| 1 Cube | $93 \pm 3$ | $36.7 \pm 2.7$ | $89 \pm 2$ | $88.7 \pm 3$ | $\mathbf{94.8 \pm 1.5}$ | $\mathbf{92.7 \pm 4.5}$ |
| 2 Cubes | $5.2 \pm 1$ | $1.3 \pm 1$ | $14.6 \pm 12.5$ | $38.8 \pm 10.6$ | $\mathbf{91.7 \pm 3}$ | $74 \pm 4$ |
| 3 Cubes | $0.6 \pm 0.1$ | $0.2 \pm 0.4$ | $14 \pm 16.4$ | $66.8 \pm 17$ | $\mathbf{89.4 \pm 2.5}$ | $62.1 \pm 4.4$ |

Table 12: Performance results on `PandaPush`, a physics-based tabletop benchmark in IsaacGym where a Franka arm must arrange multiple cubes to match target goal images. Reported values are success rates over 500 trajectories across 5 seeds; results within one standard deviation of the best are shown in bold.

GCIVL (Park et al., 2023), QRL (Wang et al., 2023), CRL (Eysenbach et al., 2022) and HIQL (Park et al., 2023).

Our image-goal-conditioned LPWM is trained on offline data with a 30-frame horizon, where goals are sampled within a window spanning the last training frame to 70 steps into the future, typically encompassing 2–3 atomic tasks. Because trajectories stem from unstructured play data rather than task-specific demonstrations, the goal sampling window is limited to maintain informative transitions for goal conditioning. During inference, we sample 20 actions per step, execute them in the environment, and feed back new observations for subsequent action predictions. Table 13 summarizes results across tasks, while Figure 4 visualizes an example imagined trajectory alongside environment execution. Videos are available on our project webpage.

*Results discussion:* OGBench datasets contain highly suboptimal, unstructured trajectories, posing challenges for behavioral cloning (BC), particularly on tasks requiring many atomic subtasks (e.g., unlock drawer, open drawer, place cube). As reflected by GCBC's performance, straightforward BC struggles when the goal is distant from the initial state. Nonetheless, our BC method achieves strong performance on tasks involving up to four atomic behaviors, including `task1` and `task3`, outperforming all baselines on these. We attribute this to LPWM's expressiveness, which captures multiple behavior modes and highlights its potential for integration with RL value functions to optimize goal-reaching policies.

`PandaPush`: designed to challenge complex, goal-conditioned multi-object manipulation. We use the same 1–3 cube manipulation dataset as EC-Diffuser (Qi et al., 2025), but unlike EC-Diffuser, we train a single multi-view image-goal-conditioned LPWM and policy across all tasks, rather than separate policies for each task (e.g., one for 1 cube, another for 2 cubes), which gives the baselines an advantage. Baselines, taken from Qi et al. (2025), include VQ-BeT (Lee et al., 2024), a non-diffusion method using a Transformer with flattened VQ-VAE image inputs; Diffuser (Janner et al., 2022), trained without guidance on flattened VQ-VAE inputs; EIT+BC, an adaptation of the EIT policy (Haramati et al., 2024) to behavioral cloning using pre-trained DLP image representations; and EC Diffusion Policy, inspired by Chi et al. (2023) and modified for goal-conditioning, learning from pre-trained DLP representations. Table 12 summarizes results, and Figure 17 shows an example imagined trajectory alongside environment execution. Videos are available on our project webpage.

*Results discussion:* despite a relatively simple policy compared to complex diffusion-based methods, LPWM outperforms all baselines except EC Diffuser and matches EC Diffuser's performance on the 1-cube task. While this work focuses on demonstrating the potential of adapting pre-trained LPWM for downstream decision-making, future work can explore combining LPWM with more advanced policies for multi-object reasoning. Additionally, we leverage the multi-view variant of LPWM in this experiment, modeling particle dynamics simultaneously from multiple views, demonstrating the framework's flexibility and enhancing the its ability to robustly handle occlusions (Haramati et al., 2024).

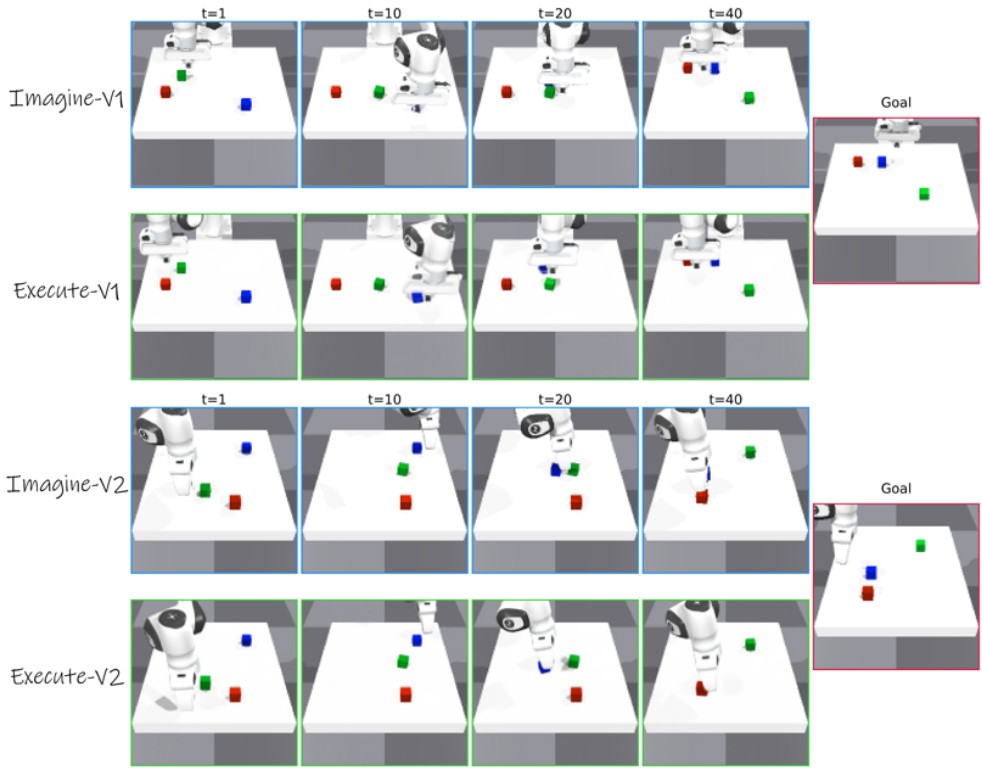

Figure 17: LPWM generated goal-conditioned imagined trajectories (top) and actual environment executions (bottom) through a learned mapping to actions on `PandaPush` from two views. LPWM generates dynamics in both views simultaneously, handling occlusions by the gripper.

| Task | GCBC | GCIVL | GCIQL | QRL | CRL | HIQL | LPWM (Ours) |
|---|---|---|---|---|---|---|---|
| task1 | $59_{\pm7}$ | $84_{\pm4}$ | $56_{\pm4}$ | $44_{\pm6}$ | $52_{\pm6}$ | $80_{\pm6}$ | $\mathbf{100}_{\pm0}$ |
| task2 | $0_{\pm0}$ | $24_{\pm8}$ | $1_{\pm1}$ | $2_{\pm2}$ | $1_{\pm1}$ | $\mathbf{81}_{\pm7}$ | $6_{\pm9}$ |
| task3 | $0_{\pm0}$ | $16_{\pm8}$ | $0_{\pm0}$ | $0_{\pm0}$ | $0_{\pm0}$ | $61_{\pm11}$ | $\mathbf{89}_{\pm9}$ |
| task4 | $2_{\pm1}$ | $0_{\pm0}$ | $3_{\pm4}$ | $2_{\pm1}$ | $1_{\pm1}$ | $\mathbf{20}_{\pm8}$ | $3_{\pm5}$ |
| task5 | $0_{\pm0}$ | $0_{\pm0}$ | $0_{\pm0}$ | $0_{\pm0}$ | $0_{\pm0}$ | $\mathbf{3}_{\pm2}$ | $0_{\pm0}$ |
| overall | $12_{\pm2}$ | $25_{\pm3}$ | $12_{\pm2}$ | $10_{\pm1}$ | $11_{\pm2}$ | $\mathbf{49}_{\pm4}$ | $40_{\pm1}$ |

Table 13: **Full results on Visual Scene** with the `visual-scene-play-v0` dataset. Results for baselines were taken from the original OGBench benchmark (Park et al., 2025) and represent success rates across 4 seeds. Results within a standard deviation are highlighted in bold.

