# OpenReview forum: "Latent Particle World Models: Self-supervised Object-centric Stochastic Dynamics Modeling"
_ICLR.cc/2026/Conference — ICLR 2026 Oral_

### Official Review · Reviewer_gkk7 · 2025-10-22

**Soundness:** 3
**Presentation:** 2
**Contribution:** 3
**Rating:** 6
**Confidence:** 4

**Summary:**

The paper proposes a novel particle based architecture for learning world models, action-conditioned video generation models. In addition to previous Particle based video models, the authors propose to combine these with per-particle latent actions, adding additional flexibility to the model.

Reviewer positionality: while I have co-authored related work, I do not normally review vision-centric papers, but RL papers. This will bias my review and I invite the authors to correct me on field-specific standards.

**Strengths:**

The authors thoroughly motivate their work and propose a reasonable addition to particle-based generative dynamics models.

The authors thoroughly ablate the base model changes as well as the proposed per-latent particle, which makes the evaluation of the method admirably robust. I have some minor questions on some of the scores, which I believe are due to my unfamiliarity with the datasets.

While the writing could be improved (see below) the overall method is straightforward and clear, and the

**Weaknesses:**

The gains on the vision datasets seem very marginal, and not significant at a reasonable confidence interval. Provided confidence intervals seem to overlap, that makes the results very hard to judge.

No information is provided on how confidence intervals (the +/- numbers in the tables) were computed. This makes the previous issue more problematic to assess.

The robotics experiments do not seem to compare planning with other latent action methods with the proposed method. That limits the clarity and scientific impact of the comparison. Can the authors provide some insights into the performance of e.g. DDLP on this task?

The robotics experiments are limited in scope and not performed on benchmarks where object interaction is particularly crucial, or where objects have complex dynamics. As one of the main motivation for learning latent actions is decision making and imitation, I think expanding the experimental section here and adding more details on the setup would greatly increase the relevance of this paper to the community. For example, while the method performs well on tasks 1 and 3, it completely fails on task 2, but what this signifies is not discussed. Furthermore, results presented in the main paper are cherry picked without justification, which I consider very bad practice.

Please specify the exact tasks chosen in OG-Bench for ease of comparison with other works.

Table 7 compares the changes to DLP introduced in this paper. The proposed method (evaluated in Table 8 on the same dataset) seems to strongly underperform this baseline, yet this is not mentioned in the ablations. I assume this is because one is tested as an image model and one as a video model, but without this context the results are confusing.

Another issue with the paper is that it only formally fulfills the criteria of being 9 pages long. The method is de-facto impossible to fully comprehend and re-implement with the main body of the paper, and so the appendix becomes essentially a part of the main paper. I would strongly advise the authors to revise and tighten the writing in the main paper to move important architectural details to there. For example, the introduction is quite lengthy, with a slightly superfluous aside on the human visual system, and dominated by a massive image, and some details on the datasets could easily be moved to the appendix. This is not a reason for me to recommend rejection, but it would probably improve the paper to tighten the writing.

**Questions:**

How are causal actions handled in the framework. I am not quite able to tell whether the dynamics are conditioned on all particle actions or not. If they aren't how are causal relations (a robot moves a block) modelled, and if they are, how is causal confusion resolved? Is the assumption that enough data will solve this issue?

Is the number of particles architecturally fixed? If yes, what is the principle advantage of particles over slots?

Can known action information (e.g. robotics datasets where action information is readily available) be used to improve and ground latent action learning. In the imitation experiments an action mapping is learned post-hoc, but for applied decision making researchers (robotics, RL, etc.) feeding available action information in at training time would greatly improve the applicability of the method.

Can pre-trained segmentation or feature learning models such as SAM, Dino, or V-JEPA be used to further improve the method or replace some components, to reduce the load of training on large scale video datasets.

---

> ### Author Response · Authors · 2025-11-17
>
> We thank the reviewer for the time and effort they put in writing the review, and we sincerely appreciate the reviewer’s honest disclaimer regarding their field of expertise. We find this extremely valuable as we aim to make the paper more friendly and welcoming to readers who are less familiar with the self-supervised object-centric learning field (we strived to do so in the appendix by providing extended related work covering the different types of self-supervised object-centric representations, extensive background on deep latent particles with code excerpts and detailed the model’s components). The focus of this paper is indeed the video dynamics prediction and we hope that during this rebuttal we can clarify your concerns and revise the paper accordingly. Below, we address the reviewer’s concerns and questions.
>
> **Video generation (vision) performance**: first, we would like to clarify that for the SSIM and PSNR metric, **higher is better** (upfacing arrow near the metric name), while for LPIPS and FVD **lower is better** (downfacing arrow near the metric name). PSNR, SSIM and LPIPS are standard pixel-based metrics to evaluate **reconstruction**, while FVD is used to evaluate **generation** (as for stochastic generation, we do not have a ground-truth pair, and we can only use empirical-distribution-based metrics). The provided standard deviation is computed over the per-video reconstruction errors, similarly to previous works ([[1]](https://arxiv.org/abs/2010.02054), [[2]](https://arxiv.org/abs/2306.05957)). LPIPS (lower is better) has been shown to correlate better with human perception ([[3]](https://www.scirp.org/journal/paperinformation?paperid=90911), [[4]](https://arxiv.org/abs/2210.05861)), and as such is a better judge of video quality. Our proposed method, LPWM, significantly outperforms the baselines on this metric, as also evident by our qualitative results (see videos on the [project webpage](https://sites.google.com/view/lpwm)), where baselines provide blurry rollouts, with objects deforming, while LPWM demonstrates object permanence and temporal coherence.
>
> **Imitation learning experiments**: the focus of this paper, and its position in the literature, is self-supervised object-centric video generation. In this work, we wanted to further demonstrate the potential of this type of models for decision-making. Previous work in this line of research typically discusses this potential, but rarely demonstrates any practical application ([[1]](https://arxiv.org/abs/2010.02054), [[4]](https://arxiv.org/abs/2210.05861)), or provides simple experiments such as measuring the action-prediction error ([[5]](https://arxiv.org/abs/2502.07600)), without actually evaluating success rates. In this work, we take a step further, and evaluate our proposed self-supervised object-centric model on complex multi-object tasks.
>
> Regarding object-centric baselines for imitation learning, DDLP is a deterministic video prediction model (i.e., it does not work on robotics datasets that involve actions as it relies on deterministic movement of objects, an assumption that does not hold in this case. See the limitations discussion under Section 6 of [DDLP](https://arxiv.org/abs/2306.05957), where it is noted that DDLP cannot handle objects appearing and disappearing, such as occlusions in robotic data, and [the authors note on its applicability to robotics datasets](https://openreview.net/forum?id=Wqn8zirthg&noteId=15yvLXHb2C) (“The BAIR robotic pushing dataset does not fit our framework as the transition between frames is stochastic”).  As shown in [GATSBI](https://arxiv.org/abs/2104.04275)), without additional guidance, such as action-conditioning these models do not perform well on stochastic video prediction, and they cannot be directly used for decision-making tasks such as imitation learning. In this work, we extend DDLP to make it applicable for stochastic dynamics prediction **and** propose a simple method to learn policies from these stochastic predictions. As it is not trivial to extract policies from the original DDLP, we focus our presentation on how to use the proposed model for that task.

---

> > ### Author Response · Authors · 2025-11-17
> >
> > On PandaPush, 3-cubes manipulation is considered a significantly hard task, where most non-object-centric models catastrophically fail ([[6]](https://arxiv.org/abs/2404.01220v1), [[7]](https://arxiv.org/abs/2412.18907)). The chosen environments include multi-object interaction (e.g., interaction between gripper and cube, and cube-cube interactions), and as demonstrated, LPWM is able to capture these intricate interactions, both in video prediction and in policy rollouts. In this work, we merely opted for demonstrating the potential and thus employed a rather simple policy mapping between the outputs of LPWM and the GT actions. Our chosen baselines are methods that are dedicated imitation learning/offline RL policy-learning methods, to emphasize our contribution. As detailed in Section 5.2, dedicated object-centric policy methods, such as EC-Diffuser ([[7]](https://arxiv.org/abs/2412.18907)), perform better, while our simple approach is highly competitive, demonstrating the potential. Future research can further explore using object-centric video models for downstream decision-making.
> >
> > **OGBench clarification**:
> > We would kindly like to emphasize that the goal of this paper is not proposing an imitation learning or offline RL method, but rather to demonstrate the potential of a novel video prediction model for downstream decision-making tasks. To that end, we picked two representative benchmarks: PandaPush for complex multi-object manipulation, and OGbench-Scene for long-horizon tasks. Specifically, for OGBench, we use the fixed offline dataset that is used to train offline RL methods for **all** tasks. We would like to emphasize that OGBench is a benchmark designed specifically to test offline RL algorithms, stressing challenges such as suboptimal data, stitching and long-horizon reasoning (see Table 13 for performance of the Goal-Conditioned Behavioral Cloning (GCBC) baseline as reference). The exact description of each task is detailed in the original [OGBench paper](https://arxiv.org/abs/2410.20092) in Figure 7 (we assume that at this point in time, the OGBench benchmark is well-known in the community).
> >
> > For tasks 4 and 5, all methods (including RL baselines) fail and there are no insights to be made for comparing the methods, and for space considerations we provide the full results in the appendix. **We will clearly communicate in the main text that for tasks 4 and 5, all methods, except HIQL on task 4, fail.**
> >
> > As for task 2, it requires completing 6 atomic behaviors which include 2 sets of sequentially dependent tasks (e.g., unlock drawer -> open drawer -> lock drawer). Obtaining this behavior requires a high degree of “stitching” given the suboptimal nature of the offline dataset, which RL methods are normally designed for, explaining their better performance.
> > The fact that LPWM outperforms all OGBench baselines on task 1 (2 atomic behaviors) and task 3 (4 atomic behaviors, 2 of them being sequentially dependent) highlights that the stitching capability does emerge to a certain extent, which we hypothesize is related to the object-centric factorization that facilitates compositionality.
> >
> > Finally, we would like to highlight a trait of our model: it can accurately produce long-horizon plans in imagination on the OGBench environment (we demonstrate imagination rollouts of 70 steps on the project website) . In practice, this enables executing 20-step **open-loop** plans (Appendix A.10.4) when solving the OGBench tasks while state-of-the-art model-based decision-making methods such as the [TD-MPC](https://arxiv.org/abs/2310.16828) line of work plan for up to 5 steps using Model Predictive Control (MPC). To the best of our knowledge, this level of accuracy is unprecedented in model-based decision-making from pixels.
> >
> > **Comparison with DLP, DLPv2**: we would kindly like to point out that Table 7 compares the **single-image** (not video) reconstruction performance and that DLPv3 **significantly outperforms** the previous iterations as indicated by the metrics (for PSNR and SSIM, higher is better, and for LPIPS lower is better, better aligned with human perception as mentioned above). In Table 8, we compare the video prediction performance on datasets with deterministic dynamics, where DDLP uses DLPv2 as the base particle encoder-decoder, and LPWM uses DLPv3. Overall, for this type of datasets, despite being lighter (as there is no latent action module), DDLP demonstrates excellent performance. LPWM is an extension of DDLP that makes it applicable for stochastic dynamics datasets and removes the need for tracking (DDLP first employs tracking over input frames, which limits the datasets it is applicable to train it on, as discussed in Appendix A.4.4).

---

> > > ### Author Response · Authors · 2025-11-17
> > >
> > > **Writing and presentation**: we appreciate the reviewer's high-level suggestions. We strived to make the paper as accessible as possible by including detailed methodological descriptions in the appendix to serve both curious readers and to facilitate future research (and as promised, we will publish an open-source code with clear instructions and documentation). As per ICLR guidelines, we have also used the opportunity to revise the manuscript and address your concerns, with all main changes clearly marked and summarized under the “General Comment”, at the top. We welcome any further specific recommendations you may have to improve clarity, especially in the method section. Making the work more approachable for readers less familiar with self-supervised object-centric approaches remains a top priority, and we greatly appreciate your input.
> > >
> > >
> > > **Causal actions**: both the dynamics and context modules are **causal**, and are using spatio-temporal causal transformers. As such, latent actions produced by the context module are indeed causal, and when used to condition the causal dynamics module, the causal structure is preserved. As detailed in the method section, differently from DDLP, we do not filter out particles after the encoding to preserve their identities when they are input to the context and dynamics modules (particles are filtered in the decoder), hence, we use **all** particles and their corresponding latent actions for the dynamics part, even for inactive particles (particles with transparency $z_{transparency}=0$).
> > >
> > >
> > > **Particles vs. slots**: thank you for this important question, as it directly relates to the different families of self-supervised object-centric learning. This question has been addressed before in DDLP ([[2]](https://arxiv.org/abs/2306.05957), Appendix C, P .23 and Appendix H, P. 35, and Table 3 that demonstrates that particles outperform slots), and we similarly addressed it in the extended related work in Appendix A.6. We summarize the main differences below:
> > > * Particles provide an explicit disentangled decomposition to attributes, such as position and scale, as opposed to slots that separate visual attributes only to different slot, where each slot compress visual information of the same resolution (e.g., if the original input image is $64 \times 64$, then each slot is decoded to a $64 \times 64$ image). In contrast, particles only model a small part of a scene (patches/glimpses), allowing high-quality reconstruction, modeling small objects and lighter memory footprint. The explicit attribute modeling has been proven to be useful for multi-object video prediction ([[2]](https://arxiv.org/abs/2306.05957)) and RL ([[6]](https://arxiv.org/abs/2404.01220v1)).
> > > * Particles are typically very low-dimensional (the effective particle dimension ,discussed in the text, is $6 + d_{\text{obj}}$, where $d_{\text{obj}}$ is the latent visual features dimension, typical values are between $4-8$. Slots are typically much higher dimensional ($64-256$) as they need to compress more information (the input to be compressed has the same resolution as the original image). This enables using a large number of particles (e.g. 64) to model a large number of objects, as opposed to slots where standard models use around 10 slots.
> > > * Slot-based models are notoriously unstable to train, and training with the same hyper-parameters may lead to inconsistent decompositions, blurry predictions, and convergence issues. This has been discussed in several previous work, and the current practice to output stable decomposition is to train the slot models over feature maps from large pre-trained models ([[8]](https://arxiv.org/abs/2209.14860), [[9]](https://arxiv.org/abs/2408.09162)). For DLP, we did not face any stability issues or inconsistent decompositions, other than choosing hyperparameters that prevent proper decomposition (e.g., setting the latent visual dimension of each particle $d_{\text{obj}}$ to be very small can lead to deferring objects to the background).
> > > * Typical slot-based models for video prediction/world modeling require a 2-stage training scheme, where first the slot decomposition is learned and then the dynamics, while DDLP and LPWM both train end-to-end (the particle decomposition is learned as part of the video prediction task).
> > >
> > > Regarding the number of particles: the maximal number of particles is fixed and is a hyperparameter. However, particles have an explicit transparency attribute, and empirically, most of the particles are inactive (transparency of 0). However, for dynamics modeling, we need to maintain the fixed number to preserve the particle identities (another option is using tracking, as in DDLP, but this has several limitations as discussed in Appendix A.4.4).

---

> > > > ### Author Response · Authors · 2025-11-17
> > > >
> > > > **Known action information**: LPWM supports action conditioning, and indeed, as the reviewer points out, it leads to better video prediction performance (see results on LanguageTable with and without action conditioning in Table 2 and 10) as the GT global actions ground the latent actions. We certainly believe that integrating LPWM in actor-critic model-based frameworks is a promising future work.
> > > >
> > > > **Usage of large pre-trained models**: using large pre-trained foundation models has recently become mainstream, and has also recently propagated to object-centric learning ([[8]](https://arxiv.org/abs/2209.14860)), mainly to mitigate stability issues in decomposition. While it would be interesting to try combining particles with, e.g., DINO, we believe it would be more interesting to scale up the particle models to serve as the large pre-trained model (reviewer yZTp has also expressed interest in that). While this work presents a big step towards this direction by demonstrating that the self-supervised object-centric DLP can directly work on various real-world datasets from scratch without the help of pre-trained models, there is still work to be done in this space to make this line of work applicable for large-scale general-purpose datasets. In addition, using these large pre-trained modules incurs a large memory footprint for inference, as they are large models (for training, one can pre-process the representation, but for inference, one still needs the models loaded in memory). As for using pre-trained supervised segmentation modules such as SAM (typically requires some sort of prompting to choose the objects), the main goal for the self-supervised object-centric community is to learn how to discover objects without the aid of supervision, and as such these models are not directly helpful.
> > > >
> > > > As for self-supervised models that do not use pixel-based reconstruction for their objective (such as V-JEPA), we believe it is a valuable line of research in the self-supervised learning community, and we are excited to see object-centric learning combined with these approaches.

---

### Official Review · Reviewer_yZTp · 2025-10-31

**Soundness:** 3
**Presentation:** 3
**Contribution:** 3
**Rating:** 8
**Confidence:** 4

**Summary:**

The paper tackles the problem of conditional world modelling with latent action discovery. The main contribution of the work is the extension of the DDLP approach to handle per-particle action latents. This gives the model more flexibility in modelling complex scenes and shows SoTA video modelling results.

**Strengths:**

* Introducing per-particle action latent, rather than considering a single global action latent (as done in past works) is very sensible, especially for complex datasets, as it introduces an additional degree of freedom and increases the representational power of the model.
* The idea of the context module that combines external conditioning with implicit actions is sound and effective. This provides a universal way to implement action conditioning.
* I find the interplay between the inverse dynamics and the latent policy modules very elegant. The learned dynamics learns the latent action space, while the policy learns the prior, and at the same time regularizes the training of the former module.
* SoTA results on video modelling

**Weaknesses:**

* The novelty is somewhat limited - the authors extend the previous work by intruding per-particle action latent. Nevertheless, it has shown improved results.
* I feel like the choice to keep all M particles (and not to perform filtering to avoid tracking) sacrifices the ability to separate real objects from “empty” slots, and thus sacrificing interpretability and explicit object modelling - which in my opinion is a nice property of the original particle models.

**Questions:**

What would it take to scale this method to work on any dataset zero-shot?

---

> ### Author Response · Authors · 2025-11-17
>
> Dear reviewer,
> we sincerely appreciate the time you invested in our work and for your knowledgeable review. We are grateful that your "Strengths" section accurately points out the novelty of our work regarding the per-particle latent-action modeling, and the elegance in training it end-to-end with the dynamics module by using the latent policy as both regularization at train time and stochastic generation at inference time. We also appreciate that you noticed that the context module can be used as a universal conditioning mechanism by mapping different conditioning modalities to per-particle latent actions.
>
> **Novelty**: as the reviewer adequately noted, the main novelty lies in the introduction of the per-particle context module (as opposed to the standard global latent-action modeling), where we propose a novel **latent policy** that is used both as **regularization** (as opposed to the fixed prior or discrete codebooks used in previous latent action methods) and for **stochastic generation**. In addition, this contribution goes **beyond the object-centric approach**, as we introduced a similar context module mechanism for DVAE (our non-object-centric patch-based baseline), making it applicable for **general purpose** methods that use the standard patch-based representation. We would like to further emphasize our contribution and novelty. First, we improve the original DLP framework (applicable for learning object-centric decomposition from single images, i.e., videos are not required) and for space considerations we detail the changes in Appendix A.4 and compare it to previous iterations of DLP (see Table 7). For particle-based dynamics prediction, we remove the tracking component of DDLP, which required critical design modifications (introduction of positional embeddings and modification of the loss function to account for the particles’ transparencies) as we detail in Appendix A.4.4. Finally, we would to kindly point out that LPWM is the **first self-supervised object-centric model that is trained end-to-end, works on real-world datasets, supports multiple types of conditioning, and can be trained with multi-view inputs**. There is no other model of this kind that supports all of these to the best of our knowledge.
>
>
> **Particle filtering**: we appreciate this comment as it emphasizes the trade-off between tracking a small set of particles and preserving the identities of *all particles*. As indicated by our results (Table 8), DDLP, a lighter model than LPWM (as there is no need for the context module), that when the setting falls under its underlying assumptions (deterministic dynamics and moderate frame rate), performs well on video prediction. However, as we detail in Appendix A.4.4, tracking has several limitations that constrains DDLP to certain types of datasets. The transition to the *particle-grid* regime allows more flexibility in modeling stochasticity (such as occlusions, appearance of new objects, stochastic movements of entities) at the cost of keeping more particles in memory (only for the context and dynamics modules, particles are still filtered in the decoder as many of them have transparency of 0) and having the context module model the per-particle latent actions. However, we would like to emphasize that **all particle attributes are maintained**, including their position (keypoint), transparency, scale and segmentation maps, as can be seen in Figure 1 and on the videos on the project website. **Visible objects can still be separated by filtering out all the particles with zero transparency** (a feature that is used for downstream tasks, such as policy learning, [[1]](https://arxiv.org/abs/2404.01220v1), [[2]](https://arxiv.org/abs/2412.18907)). As detailed in Appendix A.4.4, this design aims to combine the generality of patch-based models with the expressivity and interpretability of object-centric particles.

---

> > ### Author Response · Authors · 2025-11-17
> >
> > **Scaling**: thank you for the relevant question, we briefly discussed it in the Conclusion section and appreciate the opportunity to discuss it further. We define two aspects of *scaling*:
> > **Scaling up to large-scale data** - this type of scaling entails simply training with more data of similar type (e.g., train a single model on the combined datasets used in this work, or similar datasets from the [Open-X dataset](https://arxiv.org/abs/2310.08864)), where the videos exhibit small camera motion and recurring scenarios. In that case, we believe that scaling up the model size (e.g., more layers, heads, larger transformer inner dimension etc.) and increasing the latent visual dimension ($d_{\text{obj}}, d_{\text{bg}}$) should suffice to allow generalization; however, this would require much more computational resources than used in this work, and while we find this an interesting research direction, our resources currently prevent us from pursuing this direction.
> >
> > **Scaling up to general-purpose data** - this type of scaling entails training LPWM on general-purpose datasets such as the datasets used to train the large video generation models (e.g., YouTube data), where the data includes diverse videos without recurring scenarios.
> > The field has been actively working towards bringing object-centric models to that capability ([[3]](https://arxiv.org/abs/2209.14860), [[4]](https://arxiv.org/abs/2408.09162)), and we are excited to see where the particle models fit in that case. We believe this is a research problem fit for academia as datasets like [Something-Something-V2](https://www.qualcomm.com/developer/software/something-something-v-2-dataset) are of moderate size and can be used for training with moderate resources.

---

### Official Review · Reviewer_NJH6 · 2025-11-01

**Soundness:** 3
**Presentation:** 4
**Contribution:** 3
**Rating:** 8
**Confidence:** 4

**Summary:**

This paper introduces Latent Particle World Model (LPWM), an extension of Deep Latent Particles (DLP) for self-supervised object-centric world modeling in videos. LPWM decomposes scenes into latent particles (keypoints with attributes like position, scale, depth, transparency, and features) and models stochastic dynamics using a novel CONTEXT module that infers per-particle latent actions via inverse dynamics and a learned policy prior. The model is trained end-to-end as a temporal VAE, supporting conditioning on actions, language, images, and multi-view inputs. Evaluations focus on video prediction/generation across synthetic and real-world datasets (e.g., OBJ3D, BAIR, Mario), where it outperforms baselines in metrics like LPIPS and FVD. Additionally, LPWM is applied to goal-conditioned imitation learning on PandaPush and OGBench-Scene, showing competitive results.

**Strengths:**

1. The idea of incorporating object-centric methods into world model is well-motivated. The proposed method of flexible and supports diverse conditioning (actions, language, goals, multi-view), which is practical.

2. The authors provided extensive experimental results to show the effectiveness of the proposed LPWM. Results on real-world datasets (e.g., BAIR, Bridge) highlight its robustness beyond simulated environments.
3. LPWM achieves SOTA on stochastic video prediction (e.g., FVD of 85.45 on Sketchy vs. baselines) and competitive imitation learning performance (e.g., outperforming HIQL on OGBench task3). Ablations validate key design choices (per-particle vs. global actions, latent dimensions). The compact model size and efficiency (e.g., matching larger diffusion models on BAIR-64) underscore the benefits of inductive biases over pure scaling.

4. Particle decompositions (keypoints, masks, bounding boxes) provide inherent explainability, aligning with neuroscience inspirations (e.g., "what-where" pathways), which could appeal to applications in robotics or microscopy.

5. The supplementary material of this work is sufficient and comprehensive.

**Weaknesses:**

1. The contributions are somewhat incremental, as the authors extend an existing video prediction method (DDLP) by introducing a context module for additional conditioning. However, this should not warrant rejection, given the extensive experiments demonstrating the effectiveness of these improvements.

2. The experiments could be strengthened. While the evaluations focus primarily on video prediction, the world model is intended for policy training. Comparisons with other world models (e.g., the Dreamer series and variants) in terms of sample or learning efficiency would be valuable.

3. There are also concerns regarding the hyperparameters. The model uses diverse hyperparameters for different tasks, as detailed in Table 5. This may undermine the method's generality and raise questions about hyperparameter selection. Additional ablation studies on hyperparameter sensitivity would be beneficial.

**Questions:**

1. The paper extends DDLP via the CONTEXT module for additional conditioning (actions, language, images). Could you explain how this theoretically or practically surpasses limitations of prior object-centric video prediction methods?

2. Experiments focus on video prediction, yet world models aim to support decision-making (e.g., policy training). Could you compare LPWM to the Dreamer series or other world models on sample or learning efficiency? Alternatively, justify why the current evaluation suffices for decision-making potential.

3. Table 5 shows varied hyperparameters across tasks, potentially limiting generality. Could you explain their selection and provide ablation results on sensitivity (e.g., to latent action dimension or learning rate)?

---

> ### Author Response · Authors · 2025-11-17
>
> We thank the reviewer for their effort in writing a comprehensive review, and we are grateful for emphasizing the advantage of inductive bias over pure scaling. In the following, we address your concerns.
>
>
> **Novelty**: we would like to further emphasize our contributions. The main novelty, as indicated by the reviewer, lies in the introduction of the per-particle context module (as opposed to the standard global latent-action modeling), where we propose a novel **latent policy** that is used both as **regularization** (as opposed to the fixed prior or discrete codebooks used in previous latent action methods) and for **stochastic generation**. In addition, this contribution goes **beyond the object-centric approach**, as we introduced a similar context module mechanism for DVAE (our non-object-centric patch-based baseline), making it applicable for **general purpose** methods that use the standard patch-based representation. In addition, we improve the original DLP framework (applicable for single-image training as well) and for space considerations, we detail the changes in Appendix A.4 and compare it to previous iterations of DLP (see Table 7). For particle-based dynamics prediction, we remove the tracking component of DDLP, which required critical non-trivial design modifications, such as positional embeddings and modification of the loss function to account for the particles’ transparencies, as we detail in Appendix A.4.4. Finally, we would like to kindly point out that LPWM is the **first self-supervised object-centric model that is trained end-to-end, works on real-world datasets, supports multiple types of conditioning, and can be trained with multi-view inputs**. There is no other model of this kind that supports all of these to the best of our knowledge.
>
> **Dreamer and decision-making applications**: the Dreamer line of work ([[1]](https://arxiv.org/abs/1912.01603), [[2]](https://arxiv.org/abs/2010.02193), [[3]](https://arxiv.org/abs/2301.04104), [[4]](https://arxiv.org/abs/1811.04551)) present an **actor-critic model-based reinforcement learning approach**, where one of the components is a recurrent VAE-based dynamics model that predicts **future states and rewards based on past states and actions**. We refer to Dreamer as the complete system that includes encoder, decoder, dynamics module, actor network and critic network. The focus of this work is video prediction/generation, and our **DVAE baseline reflects the non-object-centric dynamics module akin to the one used in Dreamer**, except for replacing the RNN of the original Dreamer with a Transformer-based dynamics module, which has been shown to be superior over Dreamer’s RNN ([[5]](https://arxiv.org/abs/2310.09615)). We are certainly excited about the idea of integrating our proposed dynamics module within the model-based RL framework; however, this involves non-trivial modifications, such as reward prediction and integration with actor-critic networks, which we leave for future work.
>
> In this work, we demonstrate the potential of our world model for decision-making by applying it to **imitation learning** tasks that do not involve interaction with the environment at train time, but only at inference time, for evaluating the policy (there is no reward learning involved in the process). We use two different simulation environments and tasks (OGBench scene and IsaacGym PandaPush). Our approach involves training a simple policy mapping of latent actions from a frozen pre-trained LPWM (trained **without** actions) to ground-truth actions. While our policy design and approach are simple, our method demonstrates accurate long-horizon prediction abilities and shows competitive results compared to dedicated policy-learning approaches. This highlights the potential in leveraging our world model for sequential decision-making. We believe that further improvements are possible, e.g., with more complex policy design, and we leave it for future work.

---

> ### Author Response · Authors · 2025-11-17
>
> **Context model**: we detail the limitations the context module aims to overcome in Section 4 of the main text and Appendix A.4.3, and we are happy to clarify the contribution of the context module here: the context module is designed to address the problem of **stochastic dynamics modeling in actionless videos**. Previous object-centric approaches (e.g., DDLP ([[6]](https://arxiv.org/abs/2306.05957)), G-SWM ([[7]](https://arxiv.org/abs/2010.02054))) use a single dynamics model to jointly model next-state inference and stochasticity. This approach becomes restrictive in environments with high degrees of unpredictability (as demonstrated in [GATSBI](https://arxiv.org/abs/2104.04275)), such as robotic or real-world datasets where external actions (or other signals) crucially affect the scene but are not available for the video prediction model. Our proposed context module explicitly separates the modeling of latent actions (which encapsulate the stochastic aspects) from the dynamics prediction. This disentanglement leads to a hierarchical generative process, where the context module is composed of two heads:
>
> * The **inverse dynamics** head infers per-particle latent actions from consecutive particle states and is used for training-time conditioning, which grounds the learning of transition structure.
> * The **latent policy** head outputs the distribution of latent actions given the current state, which is used to sample latent actions at inference time (when one does not have access to future states), **thus enabling stochastic generation of future frames**. In contrast to previous latent-action models that use a fixed prior or discrete codebook regularization, in LPWM, the latent policy learns the prior of latent actions, and at the same time regularizes the inverse dynamics.
> Finally, our context module design is applicable for non-object-centric models as well (as we demonstrate with DVAE), and as indicated by reviewer yZTp, it also serves as a universal conditioning mechanism where different modalities (such as global language instruction or action) are mapped to concrete per-particle latent action per timestep. The contribution of the context model is thoroughly analyzed in the paper by comparing to models with and without it, and by ablating its design (Tables 2 and 11). We hope this adds to the motivation of disentangling action modeling and dynamics prediction.
>
> **Hyperparameters**: indeed, there are various hyperparameters for LPWM; however, as can be seen in Table 5, **hyperparameters are mostly the same across image resolutions**, and in practice, there is not much tuning required. The main hyperparameters that might vary are: number of particles (if GPU memory is not a bottleneck, one can just set it to the maximal number and filter post-training, as the transparency values for each particle indicate how many particles are active), the latent visual features dimension ($d_{\text{obj}}, d_{\text{bg}}$) and the $\beta_{\text{KL}}$, which are standard VAE hyperparameters that require tuning (but in LPWM, the values are very close across datasets and are mostly affected by the input image resolution and the reconstruction loss type, MSE or LPIPS). We ablated the hyperparameters of our main contribution, which is the dimension of the latent actions (which the model is robust to), and the type of latent action modeling (global vs. per-particle, found to be critical) in Table 11. The other hyperparameters are mostly inherited from DDLP ([[6]](https://arxiv.org/abs/2306.05957)). We hope that this addresses your hyperparameters concerns, and while we believe the experiments we performed (9 different datasets) provide sound evidence, we are happy to ablate more specific hyperparameters the reviewer requests.

---

> > ### Comment · Reviewer_NJH6 · 2025-11-26
> >
> > Thank you for the comprehensive responses. They have fully resolved all of my questions.

---

### Author Response · Authors · 2025-11-17
**General comment**

We sincerely thank the reviewers for their significant effort in providing detailed, insightful, and comprehensive feedback. Your comments have been instrumental in improving our paper. We have revised the manuscript according to your suggestions; all changes are **highlighted in blue in the updated PDF**, and a summary of these changes is provided below.

* Extended the background on Deep Latent Particles (DLP), and added a reference to a figure in the appendix.
* Extended the method description, specifically, the “Encoder” and “Decoder” paragraph in the main text.
* Clarified the contribution of the Context module in Section 4: disentanglement of modeling latent actions from the dynamics prediction, and its applicability beyond object-centric models.
* Clearly stated that for task4 and task5 of OGBench, all methods fail (with the exception of HIQL attaining $20\\%$ success rate on task4).
* Added an example rollout visualization for the learned imitation learning policy to the main text (Figure 4), more videos are available on the project website.

---

### Meta-Review · Area_Chair_gHtQ · 2026-01-07

**Summary:**

The paper received strong support from three reviewers. These reviewers highlighted the method’s state-of-the-art performance in video prediction across diverse datasets, its successful application to goal-conditioned imitation learning, and the clarity of ablation studies validating key design choices.

**Reviewer Concerns:**

The authors provided thorough and technically detailed responses to all questions and weaknesses raised by the reviewers.

**Reviewer Scores:**

Reviewer NJH6 (initial 8) would maintain their score after receiving complete answers to all technical questions.
Reviewer yZTp (initial 8) would maintain or slightly reinforce their score given the clear elaboration on scalability and particle interpretability.
Reviewer gkk7 (initial 6) would likely raise their score following rebuttal.

---

### Decision · Program_Chairs · 2026-01-26

Accept (Oral)